# A non-transcriptional function of Yap regulates the DNA replication program in *Xenopus laevis*

Rodrigo Meléndez García[1†], Olivier Haccard[2†], Albert Chesneau[1], Hemalatha Narassimprakash[2], Jérôme Roger[1], Muriel Perron[1*], Kathrin Marheineke[2*], Odile Bronchain[1*]

[1]Paris-Saclay Institute of Neuroscience, CNRS, Université Paris-Saclay, CERTOFrance, Saclay, France; [2]Université Paris-Saclay, CEA, CNRS, Institute for Integrative Biology of the Cell (I2BC), Gif-sur-Yvette, France

**Abstract** In multicellular eukaryotic organisms, the initiation of DNA replication occurs asynchronously throughout S-phase according to a regulated replication timing program. Here, using *Xenopus* egg extracts, we showed that Yap (Yes-associated protein 1), a downstream effector of the Hippo signalling pathway, is required for the control of DNA replication dynamics. We found that Yap is recruited to chromatin at the start of DNA replication and identified Rif1, a major regulator of the DNA replication timing program, as a novel Yap binding protein. Furthermore, we show that either Yap or Rif1 depletion accelerates DNA replication dynamics by increasing the number of activated replication origins. In *Xenopus* embryos, using a Trim-Away approach during cleavage stages devoid of transcription, we found that either Yap or Rif1 depletion triggers an acceleration of cell divisions, suggesting a shorter S-phase by alterations of the replication program. Finally, our data show that Rif1 knockdown leads to defects in the partitioning of early versus late replication foci in retinal stem cells, as we previously showed for Yap. Altogether, our findings unveil a non-transcriptional role for Yap in regulating replication dynamics. We propose that Yap and Rif1 function as brakes to control the DNA replication program in early embryos and post-embryonic stem cells.

**\*For correspondence:**
muriel.perron@universite-paris-saclay.fr (MP);
kathrin.marheineke@i2bc.paris-saclay.fr (KM);
odile.bronchain@universite-paris-saclay.fr (OB)

[†]These authors contributed equally to this work

**Competing interest:** The authors declare that no competing interests exist.

## Editor's evaluation

The YAP protein is well-known as a major regulator of tissue growth and repair, acting as a co-factor of transcription to promote cell proliferation and de-differentiation. In *Xenopus* egg extracts and embryos, before the initiation of transcription, this manuscript now elegantly identifies a new role of YAP in DNA replication dynamics, thus slowing down cell division rate.

## Introduction

Prior to cell division, DNA must be entirely and accurately duplicated to be transmitted to the daughter cells (*Fragkos et al., 2015*). In metazoan cells, DNA replication initiates at several thousands of fairly specific sites called replication origins in a highly orchestrated manner in time and space (*Machida et al., 2005*; *Prioleau and MacAlpine, 2016*). In late mitosis and G1 phase, origins are first 'licensed' for replication by loading onto chromatin the six ORC (origin recognition complex) subunits, then Cdc6 (cell-division-cycle 6) and Cdt1 (chromatin licensing and DNA replication factor 1), and finally the MCM (mini-chromosome maintenance) 2–7 helicase complex, forming the pre-replicative complex (pre-RC, for review see *Bell and Kaguni, 2013*). Pre-RC is subsequently activated during S-phase by cyclin- and Dbf4/Drf1-dependent kinases (CDKs and DDKs), which leads to the recruitment of many

other factors, DNA unwinding and the start of DNA synthesis at origins. In eukaryotes, segments of chromosomes replicate in a timely organized manner throughout S-phase. It is now widely accepted that the genome is partitioned into different types of genomic regions of coordinated activation (*Marchal et al., 2019*). During the first half of S-phase, the early-replicating chromatin, mainly transcriptionally active and localized to central regions of the nucleus, duplicates while late replicating chromatin, spatially located at the periphery of the nucleus, awaits until the second half (*Berezney et al., 2000*; *Hiratani et al., 2010*; *Ryba et al., 2010*). This pattern of DNA replication, also called DNA replication timing (RT) program, has been found to be stable, somatically heritable, cell-type specific, and associated to a cellular phenotype. As such, the RT can be considered as an epigenetic mark and provides a specific cell state signature (*Hiratani and Gilbert, 2009*). Interestingly, a defined RT has been observed at very early stages in development, prior to the mid-blastula transition (MBT), in embryonic cells (also called blastomeres) undergoing rapid cell cycle consisting of only S/M phases that are typical of animals with external development (*Siefert et al., 2017*). Due to the absence of most transcriptional activities in these early embryos (*Newport and Kirschner, 1982*), the very rapid DNA synthesis during these cleavage divisions relies only on a stockpile of maternally supplied determinants. To date, little is known about the molecular cues that ensure faithful and complete DNA replication during early embryonic cell divisions.

Very few gene knockouts have been shown to trigger alterations in the RT program (*Dileep et al., 2015*; *Marchal et al., 2019*). Until now, the replication timing regulatory factor 1, Rif1, is one of the very few trans-acting factors whose loss of function has been found to result in major RT modifications in multicellular organisms (*Cornacchia et al., 2012*; *Yamazaki et al., 2012*). Rif1 inhibits the firing of late origins by targeting PP1 (Protein Phosphatase 1) to those origins by counteracting Cdc7/Dbf4 dependent Mcm4 phosphorylation in budding yeast and *Xenopus* egg extracts (*Alver et al., 2017*; *Davé et al., 2014*; *Hiraga et al., 2014*) and links nuclear organization to replication timing (*Cornacchia et al., 2012*; *Foti et al., 2016*; *Gnan et al., 2021*). We previously identified Yap as another factor implicated in RT control (*Cabochette et al., 2015*). Yap is a downstream effector of the Hippo signalling pathway. It was initially identified as a primary regulator of organ growth due to its action on embryonic progenitor cells (*Huang et al., 2005*; *Lian et al., 2010*; *Ramos and Camargo, 2012*; *Zhao et al., 2010*). Yap is mostly known to exert its function as a transcriptional co-activator acting via binding to the TEADs (transcriptional enhanced associated domain transcription factors) to control transcriptional programs involved in cell proliferation, differentiation, survival, and migration (*Totaro et al., 2018*; *Zhao et al., 2008*). We previously found that *yap* is specifically expressed in neural stem cells in the *Xenopus* retina and that its knockdown in these cells leads to an altered RT program, associated with a dramatic S-phase shortening (*Cabochette et al., 2015*). However, whether Yap is directly involved in DNA replication dynamics and whether it could regulate DNA replication during early embryonic divisions in the absence of transcription remained to be investigated.

We first addressed this question by taking advantage of *Xenopus* egg extracts, a cell-free system that faithfully recapitulates all steps of DNA replication (*Blow and Laskey, 2016*; *Blow and Laskey, 1986*). We and others previously found that in this system activated replication origins are spaced 5–15 kb apart and clustered in early- and late-firing groups of origins (*Blow et al., 2001*; *Herrick et al., 2000*; *Marheineke and Hyrien, 2004*). Here, we found that Yap is recruited onto chromatin at the onset of DNA replication in *Xenopus* egg extracts in a manner that is dependent on the pre-RC formation. We also showed that Yap and Rif1 co-immunoprecipitated. Furthermore, Yap depletion altered replication dynamics by increasing the number of activated origins similarly to Rif1 depletion. As previously shown in vivo for *yap* (*Cabochette et al., 2015*), we found that *rif1* is expressed in retinal stem and early progenitor cells and involved in their RT signature. Finally, targeted protein depletion at early stages of embryonic development using a Trim-Away strategy, revealed the crucial role of both Yap and Rif1 in controlling the speed of cell divisions before MBT in vivo. Altogether, our findings unveiled Yap implication in the regulation of replication origin activation and identified Rif1 as a novel partner. We propose that Yap, like Rif1, acts as a brake during replication to control the DNA replication program in early embryos and post-embryonic retinal stem cells.

## Results

### Yap is recruited to chromatin in a pre-RC-dependent manner in *Xenopus* egg extracts

Since Yap was described as a co-transcriptional factor, we wondered whether Yap was present in *Xenopus* egg extracts that are almost devoid of transcriptional activity (*Wang and Shechter, 2016*). By quantitative western blot, we found that Yap is present in S-phase egg extracts at a concentration of 11 ng/μl (169 nM, *Figure 1—figure supplement 1*). We therefore further investigated the role of Yap during S-phase in this well-characterized in vitro replication system, where upon addition of sperm DNA to egg extracts, chromatin is assembled, replication proteins are imported, recruited on chromatin and nuclei synchronously start DNA replication. Thus, this in vitro system mimics the first embryonic S-phase. To know whether Yap interacts with chromatin during S-phase, we incubated sperm nuclei in egg extracts and collected purified chromatin fractions starting from pre-RC assembly up to ongoing DNA replication. This analysis revealed that Yap recruitment onto chromatin coincided with the loading of PCNA, an indicator of the recruitment of DNA polymerases and the start of DNA synthesis (*Figure 1A*). Yap accumulation continues throughout S-phase progression. Our results also showed that Yap is recruited to chromatin after the MCM complex (Mcm2, Mcm7) (*Figure 1A*). To address whether the recruitment of Yap could be dependent on pre-RC assembly on chromatin, we added to the egg extracts recombinant geminin, an inhibitor of Cdt1 necessary for MCM loading (*McGarry and Kirschner, 1998*; *Tada et al., 2001*). As a result, the binding of Yap on chromatin was severely delayed (*Figure 1A and B*). Thus, Yap is recruited to chromatin at the start of DNA replication and its recruitment is dependent on functional pre-RC assembly in the *Xenopus* egg extract system.

### Yap depletion triggers the acceleration of DNA synthesis in egg extracts

To directly assess the role of Yap in DNA replication, we performed immunodepletion experiments. We were able to efficiently remove Yap from egg extracts (*Figure 1C*). We then used those Yap-depleted (ΔYap) or Mock-depleted (ΔMock) egg extracts to directly visualize DNA synthesis by fluorescence microscopy after incubating sperm nuclei in the presence of rhodamine-dUTP (*Figure 1D*). Quantification of rhodamine fluorescence per nucleus after Yap depletion showed a significant increase in fluorescence intensity compared to the controls (*Figure 1E and F*; mean increase of 1.5-fold in six independent experiments), whereas the number of rhodamine positive nuclei was very similar (98.8% in Mock- versus 100% in Yap-depleted extract). This suggests that Yap depletion increases DNA synthesis. We next monitored DNA replication by following $^{32}$P-dCTP incorporation into DNA. Replication reactions in Mock- or Yap-depleted extracts were stopped at various time points during S-phase and accurate $^{32}$P-incorporation into total DNA was measured by scintillation counting (*Gillespie et al., 2012*). We found in three independent experiments that Yap depletion increased absolute DNA synthesis (*Figure 1G*). To better visualize and characterize the replication dynamics, we also quantified nascent strand progression during S-phase after $^{32}$P-dCTP incorporation using alkaline electrophoresis (see one out of eight experiments in *Figure 1—figure supplement 2*). To analyse these different experiments together, we defined four different intervals of percentages of incorporation, reflecting early (from 0% to 25% incorporation), mid (from 26% to 50%), late (from 51% to 75%), and very late (from 76 to 100%) S-phase. We then calculated the ratio of $^{32}$P- incorporation between Yap- and Mock-depleted extracts for each interval. We found that Yap depletion increased DNA synthesis by an average of 1.8-fold during early S-phase, 1.7-fold during mid S-phase, 1.6-fold during late S-phase, and 1.2-fold during very late S-phase (*Figure 1H*). This increase in DNA replication after Yap depletion could result from an earlier entry into S-phase, because of a more rapid chromatin assembly, rather than an effect on DNA replication itself. However, we were able to rule out this hypothesis using our nascent strands analysis since at the very early S-phase we did not observe any precocious start of DNA synthesis after Yap depletion (*Figure 1—figure supplement 3*). Altogether, we found that Yap depletion leads to accelerated DNA synthesis, mainly during the early to mid-stages of S-phase, demonstrating that Yap negatively regulates the progression of DNA replication.

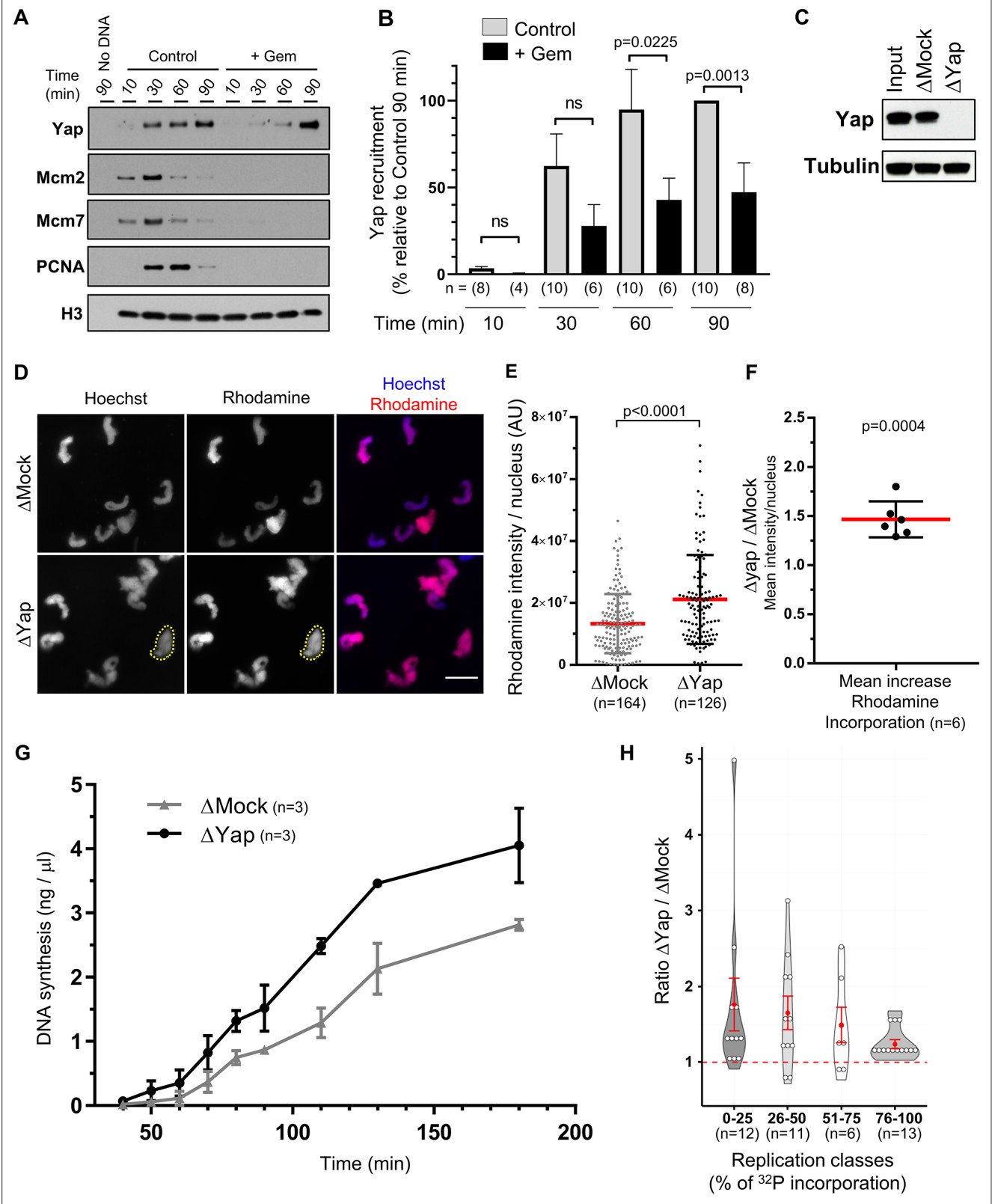

**Figure 1.** Yap is recruited to chromatin during DNA replication and the absence of Yap accelerates DNA synthesis in *Xenopus* egg extracts. (**A**) Sperm nuclei were incubated in *Xenopus* egg extracts in the absence (Control) or presence of geminin (+Gem). Chromatin was isolated for immunoblotting at indicated times points before and during DNA replication. (**B**) Quantification of chromatin-bound Yap: percentage of optical densities of the Yap bands relative to that in the control condition at 90 min in isolated chromatin fractions. The number, n, of analysed fractions per time point during DNA

*Figure 1 continued on next page*

*Figure 1 continued*

replication is indicated for each bar. Statistical differences according to Mann-Whitney test (p-values indicated; ns, not significant). Data is reported as mean ± SEM. (**C**) Western blot showing the efficiency of Yap protein depletion in *Xenopus* egg extracts. Extracts were immunodepleted with either a rabbit anti-Yap antibody (ΔYap) or a rabbit IgG as a control (ΔMock). Tubulin was used as a loading control. (**D**) Mock- or Yap-depleted extracts were supplemented with sperm nuclei and incubated with rhodamine-dUTP for 60 min. Sperm nuclei were localized by Hoechst fluorescence to define regions of interest (as exemplified for one nucleus with yellow dotted circle). Scale bar = 20 μm. (**E**) Rhodamine-dUTP incorporation was quantified as fluorescence intensity per nucleus (arbitrary units (AU); scatter blot with mean and SD; Mann-Whitney Test, two-tailed; p-value indicated). (**F**) Mean increase of fluorescence per nucleus after Yap depletion versus Mock depletion from six independent experiments (scatter blot with mean with SD; one sample t-test, two-tailed, p-value indicated). (**G**) Mock- or Yap-depleted extracts were supplemented with sperm nuclei and [α$^{32}$P]dCTP for different times, DNA was purified, counted for [$^{32}$P] incorporation and absolute DNA synthesis was calculated as ng of synthetized DNA per μg of total DNA. Means ± SEM from three independent experiments are shown. (**H**) Violin plot showing ΔYap/ΔMock ratio of incorporation from eight independent nascent DNA strands experiments. The percentage of incorporation was fractionated in four intervals to distinguish early (0–25%), mid (25–50%), late (51–75%), and very late (76–100%) phases of the replication process. The red dashed line highlights a ΔYap/ΔMock ratio of 1 that indicates no difference in the level of DNA synthesis between the two conditions, with the red dot indicating the mean and the red error bar the SEM; Wilcoxon signed ranked test, p-values: p=0.002 (0–25%, n=12), p=0.014 (26–50%, n=11), p=0.16 (51–75%, n=6), p=0.0002 (76–100%, n=13).

The online version of this article includes the following source data and figure supplement(s) for figure 1:

**Source data 1.** Related to *Figure 1*.

**Figure supplement 1.** Yap protein concentration in *Xenopus* egg extracts.

**Figure supplement 1—source data 1.** Related to *Figure 1—figure supplement 1*.

**Figure supplement 2.** Yap depletion increases nascent strand synthesis in egg extracts.

**Figure supplement 2—source data 1.** Related to *Figure 1—figure supplement 2*.

**Figure supplement 3.** Yap depletion does not affect entry into S-phase.

**Figure supplement 3—source data 1.** Related to *Figure 1—figure supplement 3*.

## Yap depletion increases replication origin firing

The higher rate of DNA synthesis observed in the absence of Yap could result from either an increase in origin firing, an increase in fork speed, or both. To directly monitor origin activation on single DNA molecules, we performed DNA combing experiments in Mock- and Yap-depleted extracts, and determined the replication content, fork density, distances between replication eyes and eye lengths (*Figure 2A–G*, *Supplementary file 1*). After Yap depletion, DNA replication significantly increased during early and mid S-phase (*Figure 2B and C*; mean increase of 2.5 times), consistent with the quantification of $^{32}$P-dCTP incorporation shown in *Figure 1G and H*. Moreover, Yap depletion significantly increased the density of active replication forks (*Figure 2D and E*; mean increase of 1.8 times), demonstrating that the absence of Yap leads to an increase in activated replication origins. In parallel, we observed a significant decrease in eye-to-eye distances (ETED; *Figure 2F*). The increase in the overall fork density was more pronounced than the decrease in distances between neighbouring origins analysed at all time points (*Supplementary file 1*). This observation highlights a role for Yap in regulating origins activation inside not-yet-active groups that replicate later. Replication eye lengths (EL) were not significantly different after Yap depletion at very early S-phase (*Figure 2G*), suggesting that fork speed remained unchanged. Larger eye sizes detected at later time points (*Supplementary file 1*) are most probably due to fusions of eyes from neighbouring origins due to increased origin activation after Yap depletion since we were not able to detect larger nascent strands in Yap depleted extracts during very early S-phase (*Figure 1—figure supplement 3*). To further confirm the role of Yap in regulating origin activation, we analysed chromatin-bound replication proteins from Mock- and Yap-depleted replication reactions in early S-phase (*Figure 2H*). We found that the initiation factor Cdc45 and elongation factor PCNA, both associated with active replication forks, were specifically enriched on chromatin after Yap depletion, consistent with an increase of active replication forks. Altogether, we conclude that Yap depletion leads to an increase in origin activation, suggesting that Yap plays a key role in limiting origin firing during DNA replication.

## Yap interacts with Rif1

To identify Yap partners in the context of DNA replication, we conducted an exploratory search for Yap-interacting proteins by co-immunoprecipitation coupled to mass spectroscopy (co-IP-MS) in S-phase egg extracts (see data availability). Among the proteins enriched more than 3-fold in Yap-co-IP versus

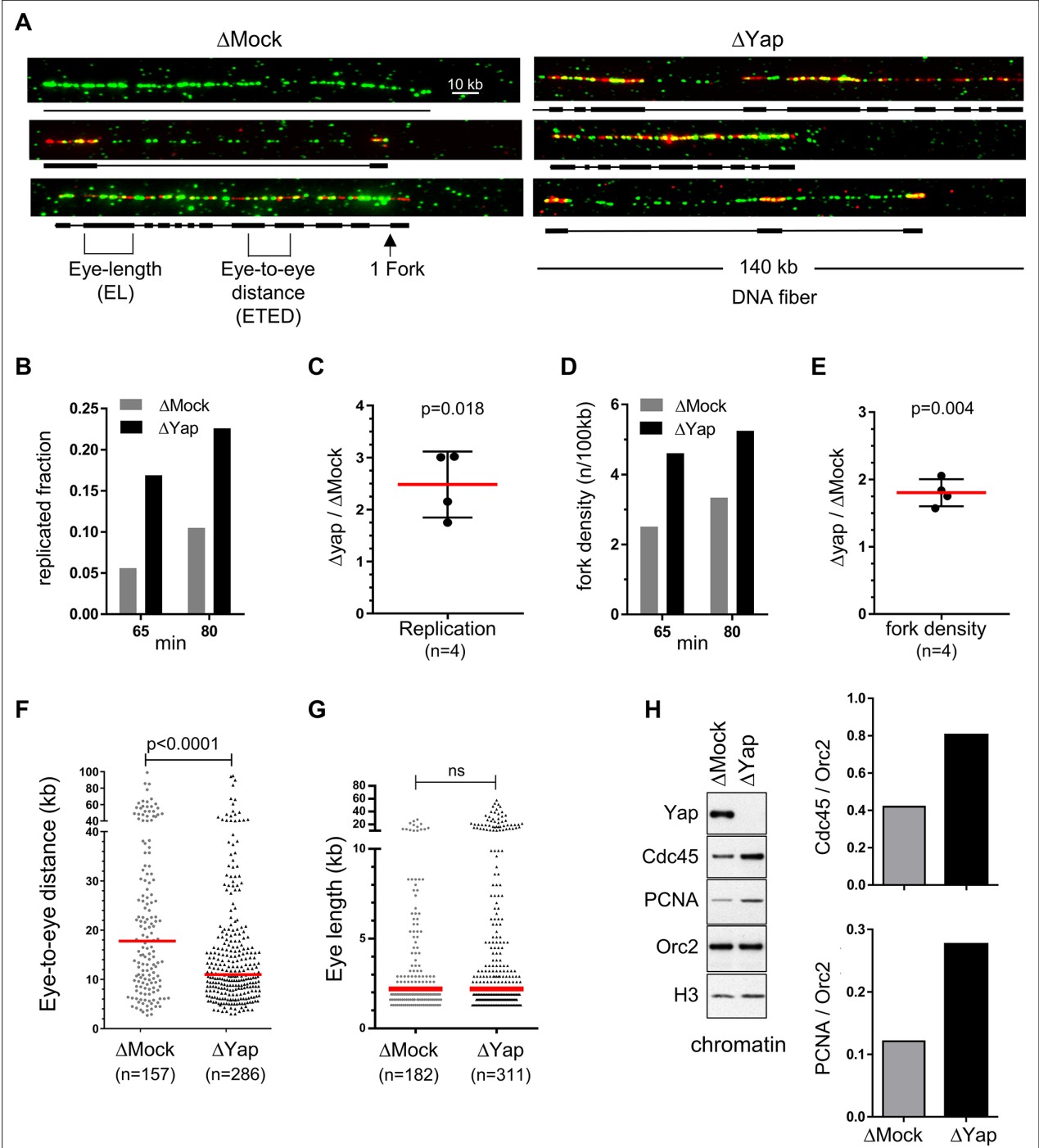

**Figure 2.** Egg extracts lacking Yap exhibit more active replication origins. Sperm nuclei were incubated in egg extracts in the presence of biotin-dUTP and DNA combing was performed. (**A**) Three representative combed DNA fibers from one combing experiment (replicate 1) after 55 min biotin-dUTP incubation in either Mock- or Yap-depleted extracts (green: whole DNA labelling; red: biotin labelled replication eyes). (**B**) Replicated fraction of one combing experiment (replicate 2) at two time points (min). (**C**) Scatter plots of ΔYap/ΔMock ratios of replicated fractions of both combing experiments at 2 time points, with mean and standard deviation, p value from one-sample t-test compared to theoretical mean 1. (**D**) Fork density (number of forks/100 kb) of one combing experiment (replicate 2) at two time points (min). (**E**) Scatter plots of ΔYap/ΔMock ratios of fork densities of both combing experiments at 2 time points, with mean and standard deviation, p value from one-sample t-test compared to theoretical mean 1. (**F**) Eye-to-eye distance (ETED) distributions of one combing experiment (replicate 2) at 80 min after Mock- or Yap-depletion, scatter dot plots with median (red bar), Mann-Whitney test. (**G**) The eye length (EL) distributions of one combing experiment (replicate 2) at 65 min after Mock- or Yap- depletion, scatter dot

*Figure 2 continued on next page*

Figure 2 continued

plots with median (red bar), Mann-Whitney test; ns, non-significant. (**H**) Left panel: western blot of chromatin bound proteins after Mock- or Yap-depletion at early S-phase with indicated antibodies. Right panel: quantification of Cdc45/Orc2 and PCNA/Orc2 ratios.

The online version of this article includes the following source data for figure 2:

**Source data 1.** Related to *Figure 2*.

Mock-co-IP conditions, we mostly identified factors functionally associated with mRNA metabolic process, ribonucleoprotein complex assembly and translation (*Figure 3A*). This is in accordance with the fact that *Xenopus* egg extracts possess little or no intrinsic transcriptional activity but can strongly support translation and post-translational modifications (*Matthews and Colman, 1991*). Of note, our analysis did not point to GO term enrichments related to DNA replication per se. However, we identified an interesting candidate, Rif1, a major regulator of the RT program (*Cornacchia et al., 2012; Yamazaki et al., 2012*). Interestingly, both Yap and Rif1 are associated with the stem cell population maintenance GO term.

We confirmed this Yap/Rif1 interaction in egg extracts by reciprocal co-IP assays (*Figure 3B*). We further validated this interaction between Rif1 and Yap following the expression of the tagged proteins in HEK293 cells (*Figure 3C*). Altogether, our data uncovered Rif1 as a Yap interacting factor, supporting the role of Yap in the regulation of DNA replication dynamics.

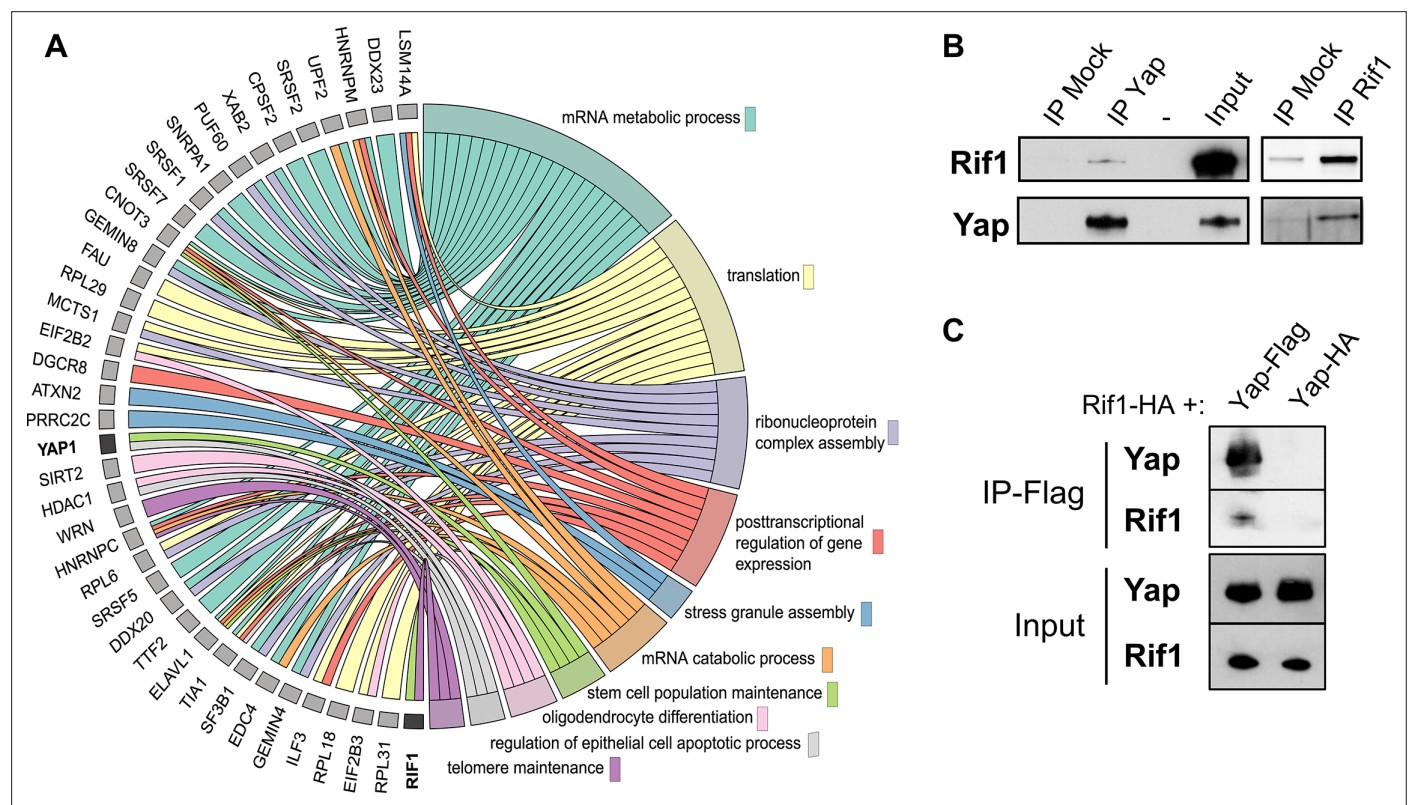

**Figure 3.** Rif1 interacts with Yap. (**A**) Chord plot representation related to GO annotations belonging to biological processes of proteins enriched by at least threefold in Yap versus control co-immunoprecipitations performed in S-phase egg extracts. Note that Yap and Rif1 are both functionally associated with stem cell population maintenance (light green). (**B**) Anti-Yap (IP Yap), anti-Rif1 (IP Rif1), or control (IP Mock) antibodies coupled to Sepharose beads were incubated in S-phase egg extracts; immunoprecipitates were subjected to gel electrophoresis and western blotted using the indicated antibodies. -, unloaded lane. (**C**) Extracts from HEK293T cells transfected with the indicated tagged constructs were immunoprecipitated using anti-Flag antibodies. The input and immunoprecipitates were subjected to gel electrophoresis and western blotted using the indicated antibodies.

The online version of this article includes the following source data for figure 3:

**Source data 1.** Related to *Figure 3*.

## Like Yap depletion, Rif1 depletion increases replication origin firing in late replication clusters

It has been shown that the depletion of Rif1 leads to an overall increase in DNA synthesis in *Xenopus* egg extracts (*Alver et al., 2017*), as we showed above for Yap depletion. However, how replication dynamics changes after Rif1 depletion in this system has not been explored. To address this and to directly compare the effects of Rif1 depletion to Yap depletion, we immunodepleted Rif1 from egg extracts (*Figure 4A*) and followed DNA replication after the incorporation of rhodamine dUTP (*Figure 4B*). We found that, as for Yap depletion (see *Figure 1D–F*), rhodamine intensity was increased after Rif1 depletion, but to a higher degree (*Figure 4C and D*). Next, we analysed origin activation after Rif1 depletion by DNA combing. We incubated sperm nuclei in the presence of biotin-dUTP for different times spanning very early to mid S-phase and isolated DNA for fiber analysis in two independent experiments (*Figure 4E*, *Supplementary file 2*). We found that Rif1 depletion strongly increased mean DNA synthesis and fork density compared to controls (*Figure 4F and G*). We further noticed that these effects were more pronounced in early S-phase compared to mid S-phase (*Figure 4H*, ΔRif1/ΔMock ratios). Therefore, Rif1 depletion led to a large increase in origin activation, especially during early S-phase. Since the percentage of unreplicated fibers decreases after Rif1 depletion compared to the control (*Supplementary file 2*), this strongly suggests that the mean 3-fold increase in active forks we observe (*Figure 4G*) is mostly due to origin activation in not-yet-activated replication clusters. We thus found that the effects of Rif1 and Yap depletions on replication dynamics are qualitatively similar since both depletions led to an early increase of entire replication cluster activations. Quantitatively, however, the mean increase of replicated fractions and fork densities were more important after Rif1 depletion (4.6-fold and 3-fold respectively), especially in early S-phase, than after Yap depletion (2.5-fold and 1.8-fold, respectively). Taken together, we conclude that Yap and Rif1 regulate replication dynamics in a similar way, likely operating through overlapping mechanisms.

## Yap and Rif1 depletions accelerate cell division rate in vivo in embryonic cells

To assess whether Yap non-transcriptional function in DNA replication dynamics also holds true in vivo, we took advantage of the early cell divisions of *Xenopus* embryos that provide a simplified system of the cell cycle. Indeed, during early development prior to the mid-blastula transition (MBT, stage 8), cells divide very rapidly, rather synchronously for a series of 12 divisions and present a cell cycle structure without gap phases. As a result, variations of the number of cells at a given time during this developmental period reflect alteration of the time spent in the S and M phases. We thus decided to deplete Yap from embryos and assess the outcomes on the rate of embryonic cell division. Since Yap protein is maternally expressed (*Figure 5A*), we employed the recently developed Trim-Away technique (*Clift et al., 2018*; *Clift et al., 2017*; *Weir et al., 2021*) to directly trigger in vivo the degradation of Yap protein stockpile. The Trim-Away-mediated knockdown has previously been shown to be effective in *Xenopus* for another target using Trim21 mRNAs (*Weir et al., 2021*). Here, we decided to use the Trim21 protein instead, to prevent delays inherent in the translation process. In addition, since we observed an increase in the level of Yap protein before MBT (*Figure 5A*), we combined the Trim-Away approach with injections of *yap* translation blocking morpholino oligonucleotides (MO) to prevent de novo protein synthesis (*Figure 5B*). We found that Yap degradation was effective from the eight-cell stage onwards using the Trim-Away approach and that the combined strategy (Trim-Away + MO) led to a stronger and prolonged Yap depletion from the eight-cell stage onwards (*Figure 5—figure supplement 1A*). We then monitored the progression of cell division before MBT. We found that Yap-depleted embryos have smaller and more numerous cells compared to controls at stage 7 (*Figure 5C*, Yap-depleted). Conversely, embryos injected with *yap* mRNA (gain of function) have larger and less numerous cells than in controls (*Figure 5C*, GOF). Importantly, the phenotype obtained after Yap-depletion was restored upon co-injection with MO-resistant *yap* mRNA (*Figure 5C*, see rescue), demonstrating specificity. Of note, the severe abnormalities observed upon Yap-depletion at the neurula stage were also rescued, further supporting the specificity of our depletion strategy (*Figure 5—figure supplement 1B,C*). Together, these data suggest that Yap is both sufficient and required to pace embryonic cleavages and that its depletion leads to an increased speed of cell divisions in pre-MBT *Xenopus* embryos.

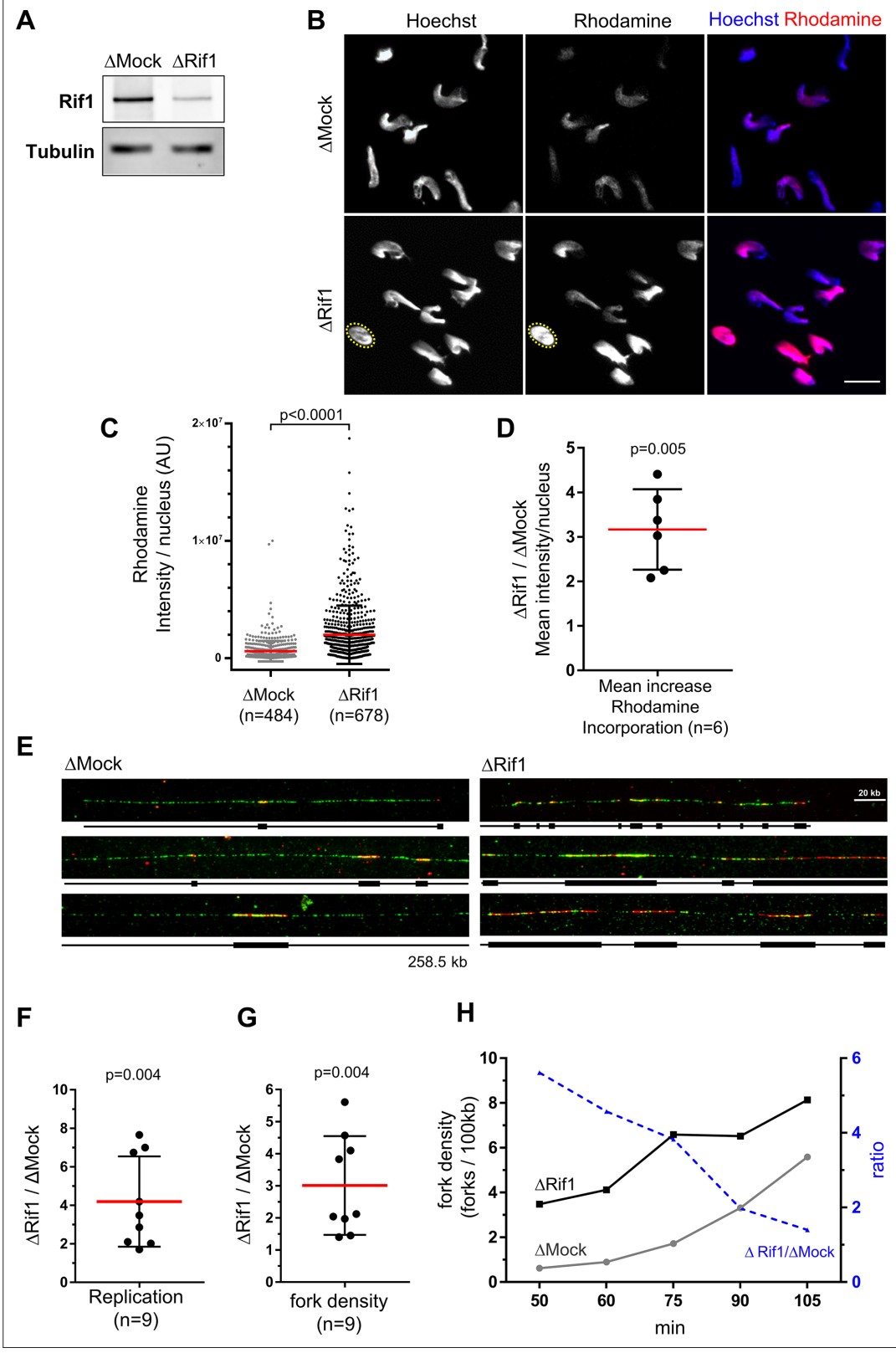

**Figure 4.** Rif1 depletion increases global fork density in egg extracts. (**A**) Western blot of egg extracts after Mock- or Rif1-depletion using the indicated antibodies. Tubulin is used as a loading control. (**B**) Mock- or Rif1-depleted extracts were incubated with sperm nuclei and rhodamine-dUTP. Replicating sperm nuclei were localized by Hoechst fluorescence (as exemplified for one nucleus with yellow dotted circle). Scale bar = 20 μm. (**C**) Rhodamine-

*Figure 4 continued on next page*

*Figure 4 continued*

dUTP incorporation was quantified as fluorescence intensity per nucleus (AU for arbitrary units), scatter plot with Mean ± SD (Mann-Whitney test, two-tailed, p-value indicated). (**D**) Mean increase of fluorescence/nucleus after Yap-depletion versus Mock-depletion from six independent experiments, scatter plot with mean ± SD, one-sample t-test (two tailed), p-value indicated. (**E**) Sperm nuclei were incubated in egg extracts in the presence of biotin-dUTP and DNA combing was performed. Three representative combed DNA fibers replicated in either the Mock- or Rif1-depleted extracts from one combing experiment (replicate 1) at 75 min (green: whole DNA labelling; red: biotin-labelled replication eyes). (**F, G**) Replicated fractions (**F**) and fork density (number of forks/100 kb) (**G**) from two independent experiments at several time periods of biotin-dUTP incorporation, Scatter plots of ΔRif1/ΔMock ratios with mean and standard deviation, p values from one-sample t-test compared to theoretical mean 1. (**H**) Fork density and fork density ratios of one combing experiment (replicate 1) at different time points (min).

The online version of this article includes the following source data for figure 4:

**Source data 1.** Related to *Figure 4*.

Since it was unknown how Rif1 depletion could affect early embryonic cell cycles in vivo, we wondered whether its depletion could lead to a phenotype similar to that obtained following Yap depletion. We undertook the same strategy to deplete Rif1 from *Xenopus* embryos using both the Trim-Away technique and *rif1*-MO. We found that Rif1 depletion from embryos also leads to an increased number of cells at stage 7, indicative of a faster rate of cell division, as observed upon Yap depletion (*Figure 5D*). Considering the known function of Rif1 in DNA replication and the absence of gap phases in pre-MBT embryos, our results strongly suggest that the increased rate of cell division in absence of Rif1 results from an acceleration of DNA replication and a shortening of S-phase length. We therefore propose that both Yap and Rif1 are involved in controlling the DNA replication dynamics in pre-MBT embryos.

## *rif1* is expressed in retinal stem cells and its knockdown affects their temporal program of DNA replication

Since we found this new interaction between Rif1 and Yap and since Rif1 has been recently shown to function in a tissue-specific manner (*Armstrong et al., 2020*), we investigated its expression and function in *Xenopus* retina and compared the results with our previous findings regarding Yap retinal expression/function (*Cabochette et al., 2015*). In situ hybridization study and immunostaining experiments revealed prominent *rif1* expression in the periphery of the ciliary marginal zone (CMZ) of the retina (*Figure 6A and B*), a region containing stem and early progenitor cells (*Perron et al., 1998*), and where *yap* is also specifically expressed (*Cabochette et al., 2015*). Double staining confirmed the co-expression of Yap and Rif1 in CMZ cells (*Figure 6C*). Of note, we verified the specificity of both Yap and Rif1 antibodies for immunostaining on retinal sections (*Figure 6—figure supplement 1*).

We next undertook a knockdown approach using *rif1*-MO (*Figure 6D*). Morphant tadpoles exhibited significantly reduced eye size compared to controls (*Figure 6E and F*), as did *yap* morphants (*Cabochette et al., 2015*). Importantly, in support of the specificity of *rif1*-MO, this phenotype was restored upon co-injection of a *rif1*-MO with MO-resistant *rif1* mRNAs (*Figure 6—figure supplement 2*).

We next analysed the level of proliferation within the CMZ in *rif1* morphant tadpoles (*Figure 7*). Unlike the observed decrease of the EdU cell number in *yap* morphant CMZ (*Cabochette et al., 2015*), we did not find any significant difference in the number of EdU$^+$ cells in *rif1*-MO-injected tadpoles compared to controls (*Figure 7A*). Interestingly, however, as observed in *yap* morphants (*Cabochette et al., 2015*), we found a drastic change in the distribution of EdU-labelled replication foci in retinal stem and early progenitor cells, where *rif1* is normally expressed (*Figure 7B–D*). Short pulse labelling experiments indeed allow the visualization of replication foci in nuclei. The spatial distribution of these foci evolves in a stereotyped manner during S-phase (*Figure 7B*): from numerous small foci located throughout the nucleus in early S-phase, to few large punctuated ones in mid/late S-phase (*Koberna et al., 2005*; *van Dierendonck et al., 1989*). Our analysis revealed a decreased proportion of cells exhibiting a mid-late versus early S-phase patterns in *rif1* morphants compared to controls (*Figure 7C and D*). We thus propose that, like *yap* knockdown, *rif1* knockdown alters the spatial organization of replication foci in CMZ cells, suggesting that both Yap and Rif1 may control the RT program in vivo in post-embryonic retinal stem/early progenitor cells.

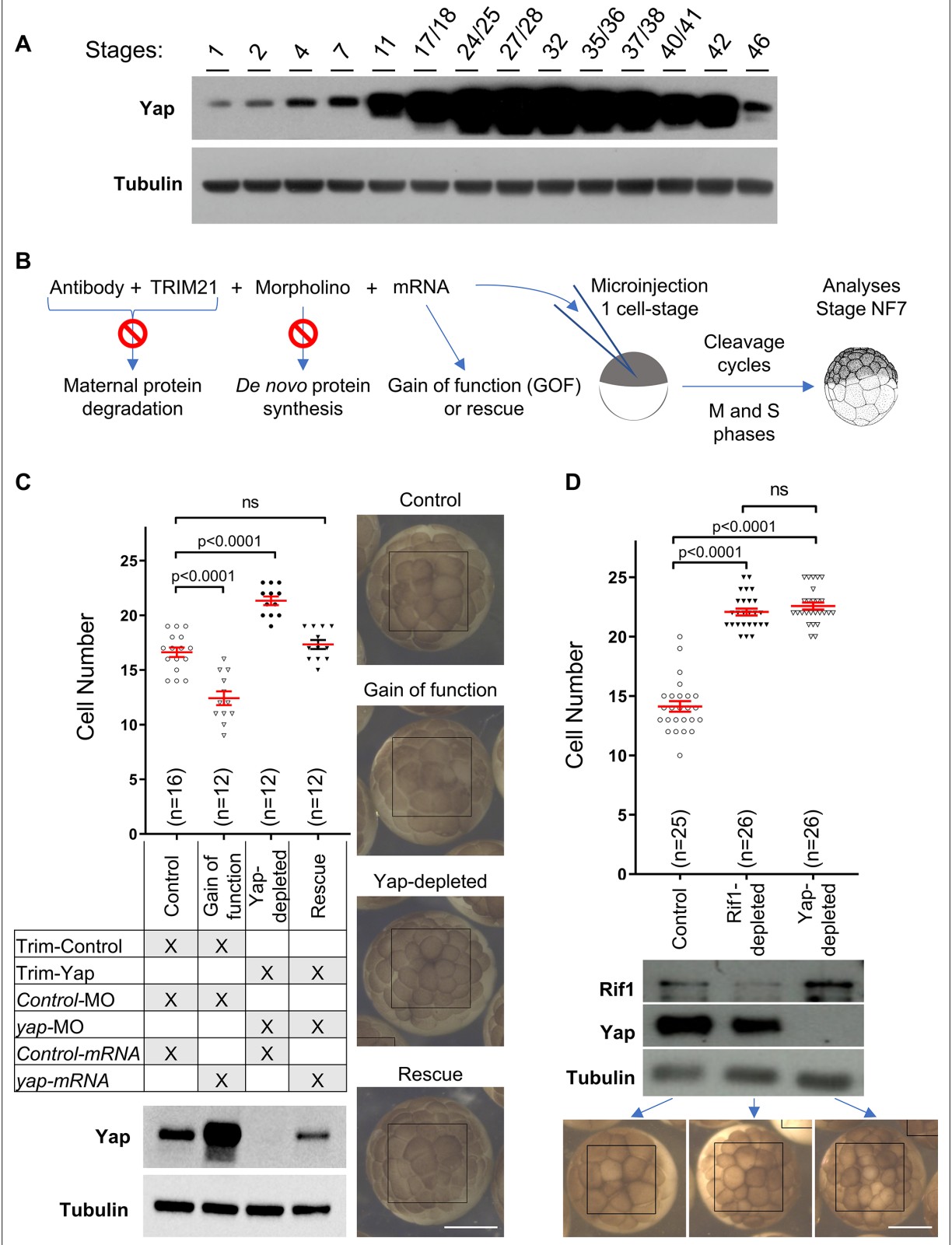

**Figure 5.** Yap and Rif1 depletions accelerate cell cycles in early *Xenopus* embryos. (**A**) Time course analysis of Yap expression throughout development by western blot. Tubulin is used as a loading control. (**B**) Diagram of the experimental procedure used in (**C**). (**C**) Evaluation of the number of cells per *Xenopus* embryo, at stage 7, following *Yap* expression perturbations. The number of cells per embryo within a defined area (black boxes on the right pictures) was quantified (top panel). Data are shown as individual value plots with error bars (mean ± SEM) in red; Mann-Whitney test; p-values

*Figure 5 continued on next page*

*Figure 5 continued*

indicated; ns, not significant. *X. laevis* embryos were microinjected at the one-cell stage as shown in the table to obtain four different levels of Yap expression: unaffected situation (control), gain of function (GOF), loss of function (Yap-depleted) and a restored expression (rescue). Yap levels of expression were monitored by western blot (bottom panel, tubulin is used as a loading control). Representative images of injected embryos are shown on the right. Scale bar = 500 μm. Trim-Control = pre-immune serum + TRIM21; Trim-Yap = anti Yap antibody + TRIM21, *Control*-MO = control morpholino (MO), *yap*-MO = morpholino targeting *yap* mRNA, *Control mRNA* = GFP mRNA, *yap-mRNA* = mRNA encoding *yap* that is non-targetable by the *yap*-MO. (**D**) Evaluation of the number of cells in *Xenopus* embryos at stage 7 within a defined area (black boxes on the bottom pictures) following Yap or Rif1 depletions. The number of cells per embryo was quantified as in (**C**). Protein depletion efficiencies were assessed by western blot (middle panel). Representative images of injected embryos are shown at the bottom. Scale bar = 500 μm.

The online version of this article includes the following source data and figure supplement(s) for figure 5:

**Source data 1.** Related to *Figure 5*.

**Figure supplement 1.** *Yap* knockdown using the Trim-Away strategy is effective at very early stages of development.

**Figure supplement 1—source data 1.** Related to *Figure 5—figure supplement 1*.

## Discussion

During S-phase, eukaryotic DNA is not replicated all at once, but large genomic regions are duplicated in a characteristic temporal order known as the RT program. To date, very few factors involved in the orchestration of this program have been identified. We have previously revealed that *yap* knockdown is associated with an altered RT program in *Xenopus* retinal stem cells (*Cabochette et al., 2015*). Whether and how Yap could directly regulate DNA replication was however unknown. Here, we used the *Xenopus* in vitro replication system and early *Xenopus* embryos, where DNA transcription is absent, to study the role of Yap in S-phase, independently from its transcriptional function. Our study shows that Yap regulates DNA replication dynamics in these embryonic systems. First, we found that Yap is recruited to chromatin at the start of DNA synthesis, and this is dependent on the pre-replication complex assembly. Second, our data revealed a non-transcriptional role for Yap in the initiation of DNA replication. Third, we identified Rif1, a global regulator of the RT program, as a novel Yap partner. Finally, our in vitro and in vivo data suggest that Yap and Rif1 are similarly involved in both DNA replication dynamics and the spatial organization of DNA replication foci. We propose a model in which Yap and Rif1 would limit replication origin firing and as such act as brakes during S-phase to control the DNA replication program in early *Xenopus* embryos and retinal stem cells (*Figure 8*).

The molecular control of the RT program remains elusive. Regarding key factors, Rif1 was the first mammalian factor shown to temporally control DNA replication, acting negatively on origin activation (*Cornacchia et al., 2012*; *Hayano et al., 2012*; *Yamazaki et al., 2013*; *Yamazaki et al., 2012*). This function of Rif1 depends on its interaction with protein phosphatase 1 (PP1) (*Davé et al., 2014*), which counteracts the DDK dependent activation of MCM2-7. On the other hand, Polo-like kinase 1 (Plk1) positively controls replication origin firing by negatively regulating Rif1-PP1 interaction (*Ciardo et al., 2021b*). Here, we found that Yap is a novel key component of this molecular machinery that regulates replication origin activation. First, we found that Yap and Rif1 physically interact. Whether this interaction could impact Rif1-PP1 association remains to be determined. Interestingly, it has previously been shown that PP1 interacts with and dephosphorylates Yap (*Wang et al., 2011*), suggesting the potential existence of a Yap-Rif1-PP1 multi-protein complex. Second, we found that Yap depletion leads to defects in DNA replication dynamics similar to those obtained following Rif1 depletion in egg extracts. Finally, we also observed similar phenotypes following *rif1* or *Yap* knockdown in *Xenopus* early embryos (i.e. increase in cell cycle speed) and in retinal stem cells (i.e. increase in early-like replication foci patterns). It is thus tempting to speculate that Rif1 and Yap act in concert to regulate replication dynamics.

We observed that Yap is recruited to replication competent chromatin at the start of S-phase then accumulates over S-phase, consistent with a direct role in DNA replication regulation. This dynamic behaviour is similar to the observed increase of chromatin bound Rif1 (*Kumar et al., 2012*). Furthermore, we showed that Yap loading on chromatin depends on functional pre-RC assembly or DNA replication per se, since inhibition of pre-RC assembly also inhibits S-phase entry. We however do not know how Yap is recruited to chromatin in the first place since our proteomic analysis did not reveal a direct interaction with any members of the pre-RC complex. Therefore, Yap might be recruited by proteins involved in steps downstream of pre-RC assembly. We found that the increased rate in DNA synthesis after Yap depletion is due to an increase in replication origin activation, especially early in S-phase, strongly suggesting that Yap directly limits origin

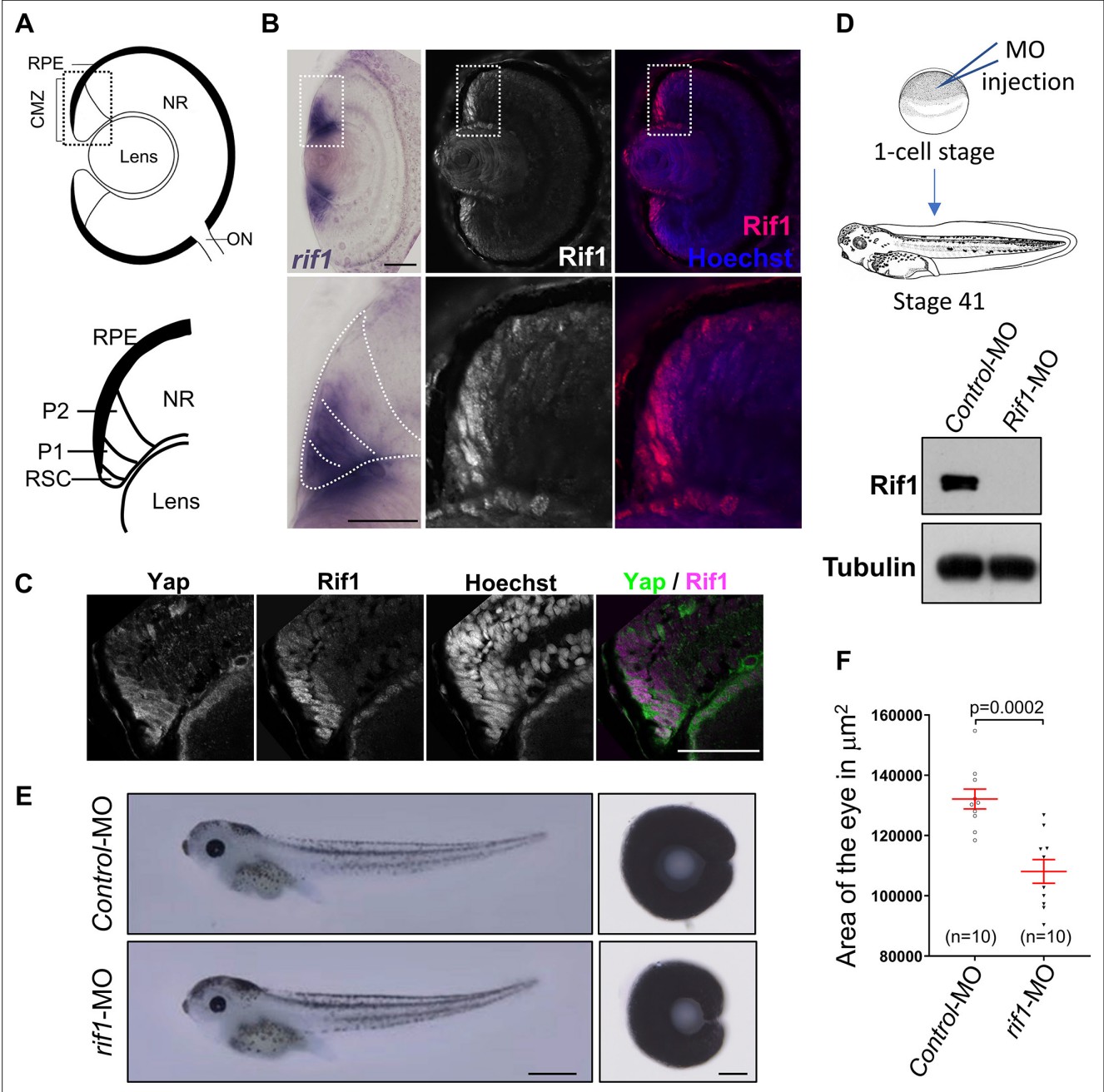

**Figure 6.** *rif1* is expressed in retinal stem cells and its knockdown leads to small eye phenotype. (**A**) Schematic transversal section of a *Xenopus* tadpole retina (RPE: retinal pigmented epithelium; NR: neural retina; ON: optic nerve). Within the ciliary marginal zone (CMZ; lower diagram), retinal stem cells (RSC) reside in the most peripheral margin, while early (P1) and late (P2) progenitors are located more centrally. (**B**) Retinal sections from stage 41 *Xenopus* tadpoles following in situ hybridization for *rif1* expression (left panels, in purple) or immunostained for Rif1 (middle panel Rif1 alone in white and right panel in red along with nuclei counterstained with Hoechst in blue). The images on the lower panels are higher magnification of the CMZ (delineated dotted lines on the top panels). (**C**) CMZ region of retinal sections from stage 41 *Xenopus* tadpoles co-immunostained for Yap, Rif1 along with nuclei counterstained with Hoechst. The left panel shows a merged picture of Yap (green) and Rif1 (Magenta). (**D**) Diagram showing the experimental procedure used in (**E**). One cell-stage embryos are microinjected with Control MO or *rif1*-MO and analysed at stage 41. The western blot shows the efficiency of the MO at depleting Rif1 in embryos. (**E**) Tadpoles microinjected with MO as shown in (**D**) and corresponding dissected eyes (right panels). (**F**) The area of dissected eyes was measured for 10 embryos per condition. Data are shown as individual value plots with error bars (mean with SEM in red; Mann-Whitney test, p-value indicated). Scale bar = 50 µm in (**B, C**), 1 mm in (E, tadpoles) and 100 µm in (E, dissected eyes).

The online version of this article includes the following source data and figure supplement(s) for figure 6:

**Source data 1.** Related to **Figure 6**.

*Figure 6 continued on next page*

*Figure 6 continued*

**Figure supplement 1.** Validation of the specificity of YAP and Rif1 antibodies.

**Figure supplement 1—source data 1.** Related to *Figure 6—figure supplement 1*.

**Figure supplement 2.** The *rif1*-MO-induced small eye phenotype is rescued by co-injection with *rif1* mRNA.

**Figure supplement 2—source data 1.** Related to *Figure 6—figure supplement 2*.

firing. Whether it prevents late origin firing in early S-phase cells or whether it inhibits dormant origin firing around active replication forks remains to be investigated. However, the increase in early-like foci patterns at the expense of late-like ones in retinal stem cells observed upon *yap* (*Cabochette et al., 2015*) or *rif1* knockdown (this study) rather suggests an impact on the partition between early and late replication-firing. In Rif1-depleted Hela cells, the overall replication foci were similarly found to be extensively rearranged, with cells displaying predominantly early S-phase-like patterns (*Yamazaki et al., 2012*). In addition, our DNA combing analysis after Yap depletion demonstrated that the overall fork density was increased to a higher extent than local origin distances were decreased. This suggests that Yap controls origin firing more at the level of groups of origins, or replication clusters, than at the level of single origins, therefore regulating the temporal control of origin activation. Similarly, we found that in Rif1-depleted egg extracts, origin activation was more increased in early S-phase compared to mid S-phase, which points toward a role of Rif1 as a replication timing factor also in early *Xenopus* embryos. Although Rif1 seems to inhibit late origin activation to a greater extent than Yap, their physical interaction and the qualitatively similar effect on fork density suggests that Rif1 and Yap act in at least partially overlapping mechanisms.

In embryos, the increased speed of cell division during early embryonic cleavage cycles that we observed upon *yap* or *rif1* knockdown is consistent with a function of both factors in limiting the replication of late genomic regions, leading to a slowing down of S-phase. In retinal stem cells, however, Yap and Rif1 do not seem to function similarly on the cell cycle kinetics. Indeed, although knockdown phenotypes were similar in terms of early-late foci ratio regulation, only *Yap* knockdown led to a significant change in the proportion of S-phase cells among stem and progenitor cells (i.e. number of EdU + cells at the tip of the CMZ). How cell cycle kinetics is differentially regulated in both cases remains to be investigated. It is well known that Yap also transcriptionally regulates cell cycle genes, which may contribute to such different outcomes. Interestingly, among direct targets genes regulated by Yap, there are also essential factors involved in replication licensing, DNA synthesis and repair (e.g. *CDC6*, *GINS1*, *MCM3*, *MCM7*, *POLA2*, *POLE3*, *TOP2A*, and *RAD18*; *Zanconato et al., 2015*). Yap may thus likely impact replication dynamics at both the transcriptional and non-transcriptional levels in retinal stem cells.

Combined observations point to a role for Rif1 in higher order chromatin architecture and its relationship with the RT program (*Foti et al., 2016*). Rif1 localizes to late-replicating sites of chromatin and acts as a remodeler of the three-dimensional (3D) genome organization and as such defines and restricts the interactions between replication-timing domains (*Foti et al., 2016*). Although the RT program can be established independently of the spatial distribution of replication foci, nuclear organization and RT are correlated and Rif1 is central in co-regulating both processes (*Gnan et al., 2021*). In this context, it should be noted that the depletion of Rif1 from egg extracts strongly increases the activation of entire replication clusters in early S-phase and this, within a system that was presumed to be physiologically at its maximum initiation rate. This raises the question of whether part of the structural role of Rif1 would be to limit the accessibility of replication factors to chromatin resulting in the observed temporal replication program. Whether Yap function in DNA replication could be linked to a role as an organizer of nuclear architecture, similar to Rif1, would thus be interesting to assess in the future. Recent studies invoke liquid-liquid phase separation (LLPS) in the establishment of chromatin activity and the formation of chromatin compartments (*Rippe, 2021*). Interestingly, Yap has been described to form liquid-like condensates in the nucleus (*Cai et al., 2019*). Whether higher level replication organization is impacted by LLPS is currently unknown, but several members of the pre-RC complex (Orc1, Cdc6, Cdt1) are also able to induce DNA dependent LLPS in vitro (*Parker et al., 2019*).

Not much is known about signalling pathways regulating the RT program or Rif1 activity. The ATM/53BP1 signalling pathway has been identified upstream of the RT and relays information onto Rif1 activity in response to DNA double-strand breaks (*Kumar and Cheok, 2014*). Future work will be required to assess whether Yap activity in the context of DNA replication is regulated by the Hippo pathway. Interestingly, LATS1, another component of the Hippo pathway, has been involved in the

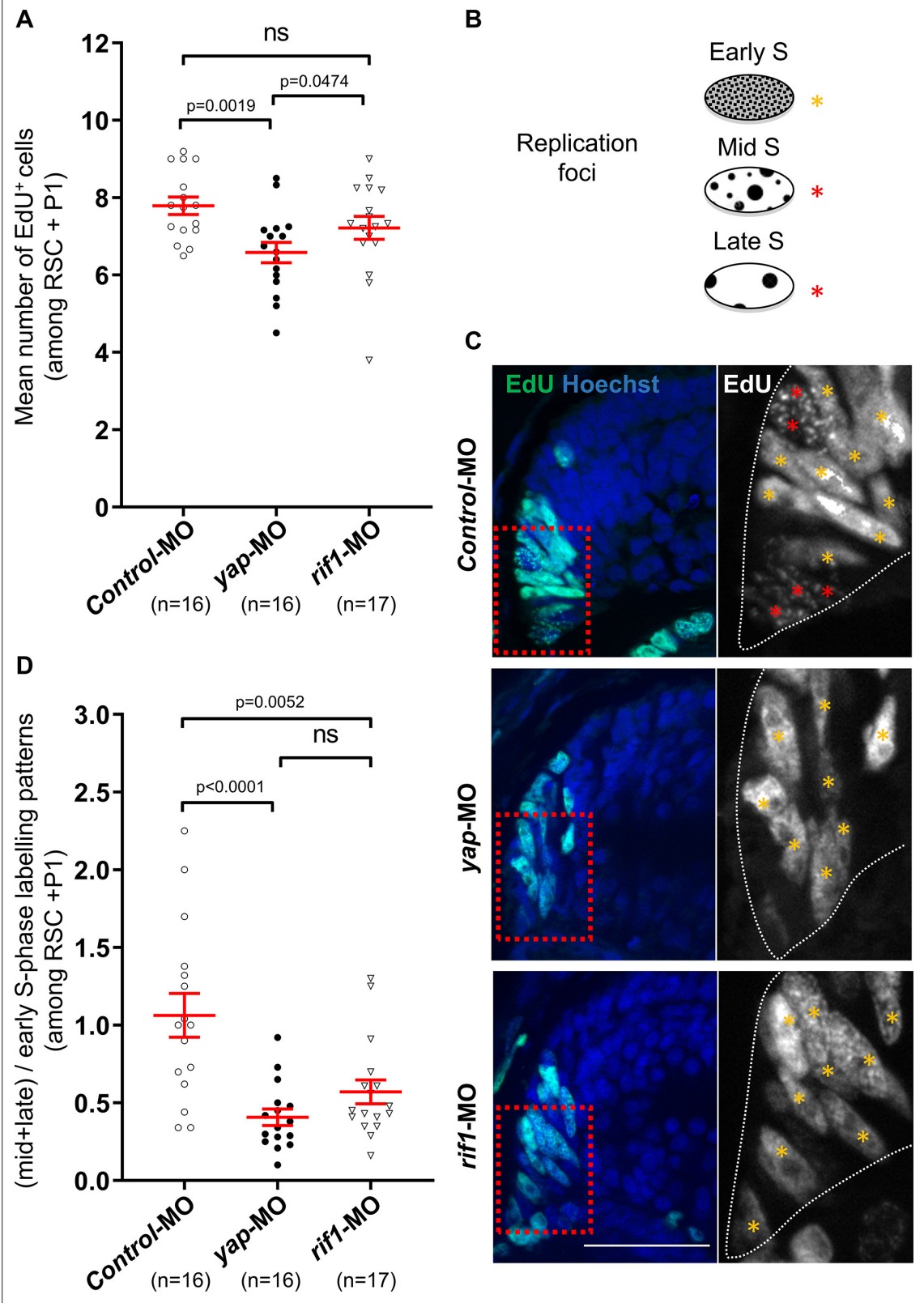

**Figure 7.** *rif1* loss of function affects DNA replication timing in retinal stem/early progenitor cells. (**A**) One-cell stage embryos were microinjected with either the control-MO (Control), *yap*-MO or *rif1*-MO and analysed for EdU-labelling (1 hr-pulse) at stage 41. Quantifications of EdU+ cell number in the retinal stem cells (RSC) and early progenitors (P1) regions (see diagram shown in *Figure 6A*). The number of analysed retinas is indicated for each condition. Data are shown as individual value plots with error bars (mean ± SEM in red; Mann-Whitney test; p-values indicated; ns, non-significant).

*Figure 7 continued on next page*

*Figure 7 continued*

(**B**) Schematic representation of the replication foci observed during S-phase progression as inferred from EdU labelling. Orange stars indicate typical early S replication patterns (homogeneous staining) while red stars indicate mid/late S replication ones (punctuated staining). (**C**) Retinal sections from stage 41 *Xenopus* tadpoles treated as described in (**A**). The region enlarged on the right panels is delineated with red dashed lined boxes. The outlines of the CMZ are highlighted by dotted white lines in the enlargements. Nuclei are counterstained with Hoechst. Scale bar = 50 µm. (**D**) Quantifications of the ratio of (mid + late)/early-like foci patterns in the retinal stem cells (RSC) and early progenitors (P1) regions. The number of analysed retinas is indicated for each condition. Data are shown as individual value plots with error bars (mean ± SEM in red; Mann-Whitney test; p-values indicated; ns, non-significant).

The online version of this article includes the following source data for figure 7:

**Source data 1.** Related to *Figure 7*.

ATR-mediated response to replication stress (*Pefani et al., 2014*). Several Hippo pathway components may thus regulate, independently or in concert, replication dynamics.

The role of Yap and Rif1 in the regulation of the RT program in early embryos opens new questions regarding the dynamics of RT changes during development. Although it was previously thought that the spatio-temporal replication program is not established until the MBT (*Hyrien et al., 1995*; *Sasaki et al., 1999*), it was also demonstrated that the oocyte-type of 5S RNA genes replicate later than the somatic-types of 5S RNA genes in *Xenopus* egg extracts (*Wolffe, 1993*). It was then shown that the RT program in *Xenopus* in vitro system is not completely random, with large chromosomal domains being replicated in a reproducible manner (*Labit et al., 2008*). Moreover, in zebrafish embryos, a genome-wide approach clearly established that a compressed, yet defined, RT program is evident before the MBT (*Siefert et al., 2017*). Further on, it was recently demonstrated that the replication program is regulated at the level of large domains by the replication checkpoint (*Ciardo et al., 2021a*) and by polo-like kinase 1 via the inhibition of Rif1/PP1 (*Ciardo et al., 2021b*). Altogether, those findings strongly suggest the existence of an embryonic RT program before the MBT. Remarkably, gradual changes in the RT program occur from the MBT and throughout development. The molecular control behind this dynamic is unknown. How Yap and Rif1 functions evolve at different stages and impact changes in the RT program at this important transition is therefore an interesting issue to be addressed in the future. To do so, the new combination of tools implemented in this study will be very valuable, as we have proven the efficient depletion of maternal proteins stockpiles while preventing their de novo synthesis by combining the Trim-Away technique with MO injections before MBT. In this context, *Xenopus* embryos seem particularly suitable to shed light and dissect the molecular mechanisms at work during embryogenesis that underlie RT changes. Identifying and characterizing the factors controlling these changes during development will certainly have an impact on how we approach questions related to cellular reprogramming, stem cells and cancer biology.

## Materials and methods

### Key resources table

| Reagent type (species) or resource | Designation | Source or reference | Identifiers | Additional information |
|---|---|---|---|---|
| Antibody | Anti-human MCM2 (rabbit polyclonal) | Bethyl lab, Euromedex, Souffelweyersheim, France | Cat# A300-191 RRID: AB_162709 | WB (1:2000) |
| Antibody | Anti-*Xenopus* MCM7 (rabbit polyclonal) | Gift from R. A. Laskey | doi: 10.1073/pnas.93.19.10189 | WB (1:1000) |
| Antibody | Anti-α Tubulin (mouse monoclonal) | Sigma, Saint-Quentin-Fallavier, France | Cat# T5168 RRID: AB_477579 | WB (1:10,000) |
| Antibody | Anti-rat PCNA (mouse monoclonal) | ThermoFisher, Illkirch, France | Cat# MA5-11358 RRID: AB_10982348 | WB (1:500) |
| Antibody | Anti-human Yap (mouse monoclonal) | Abcam Cambridge, UK | Cat# Ab56701 RRID: AB_2219140 | WB (1:1000), IHC (1:50) |
| Antibody | Anti-human Yap (rabbit polyclonal) | Abcam Cambridge, UK | Cat# Ab62752 RRID: AB_956477 | Immunodepletion (1 µl per 20 µl extract), IP (1 µl per 20 µl extract) |

*Continued on next page*

*Continued*

| Reagent type (species) or resource | Designation | Source or reference | Identifiers | Additional information |
|---|---|---|---|---|
| Antibody | Anti-human H3 (rabbit polyclonal) | Abcam Cambridge, UK | Cat# Ab1791<br>RRID: AB_302613 | WB (1:10000) |
| Antibody | Anti-Flag (rabbit polyclonal) | Cell Signaling, OZYME, Saint-Cyr-l'École, France | Cat# F7425<br>RRID: AB_439687 | IP (1 µl per test) |
| Antibody | Anti-human ssDNA (mouse monoclonal) | Merck Millipore, Guyancourt, France | Cat# MAB3034<br>RRID: AB_11212688 | DNA combing (1:50) |
| Antibody | Anti-*Xenopus* Yap (rabbit polyclonal) | This paper Covalab, Villeurbanne, France | | IHC (1:100),<br>WB (1:2000),<br>IP (1 µl per 2,5 µl extract)<br>Immunodepletion (1 µl per 2,5 µl extract),<br>Trim away (50 nl per injection) |
| Antibody | Anti-*Xenopus* Rif1(rabbit polyclonal) | This paper Covalab, Villeurbanne, France | doi: 10.1093/n10.1093/nar/gkab756 | IHC (1:100),<br>WB (1:2000),<br>IP (1 µl per 2 µl extract)<br>Immunodepletion(1 µl per 2 µl extract),<br>Trim away (50 nl per injection) |
| Antibody | Anti-Mouse IgG (rabbit polyclonal) | Sigma, Saint-Quentin-Fallavier, France | Cat# M7023<br>RRID: AB_260634 | Immunodepletion (1 µl per 2 µl extract) |
| Antibody | Anti-mouse Alexa 488 (rabbit polyclonal) | ThermoFisher, Illkirch, France | Cat# A11059<br>RRID: AB_2534106 | DNA combing (1:50)<br>IHC (1:50) |
| Antibody | Anti-rabbit Alexa 448 (goat polyclonal) | ThermoFisher, Illkirch, France | Cat# A11008<br>RRID: AB_143165 | DNA combing (1:50) |
| Antibody | Anti-mouse Alexa 594 (goat polyclonal) | ThermoFisher, Illkirch, France | Cat# A11005<br>RRID: AB_2534073 | DNA combing (1:50),<br>IHC (1:1000) |
| Antibody | Anti-mouse Alexa 488 (rabbit polyclonal) | ThermoFisher, Illkirch, France | Cat# A11001<br>RRID: AB_2534069 | IHC (1:1000) |
| Antibody | Anti-streptavidin biotinylated (goat polyclonal) | Eurobio, Les Ulis, France | Cat# BA-0500<br>RRID: AB_2336221 | DNA combing (1:50),<br>IHC (1:50) |
| Antibody | Anti-mouse IgG HRP (goat polyclonal) | Sigma, Saint-Quentin-Fallavier, France | Cat# A4416<br>RRID: AB_258167 | WB (1:10000) |
| Antibody | Anti-rabbit IgG HRP (donkey polyclonal) | GE Healthcare, France | Cat# NA934<br>RRID: AB_772206 | WB (1:10000) |
| Peptide, recombinant protein | Streptavidin Alexa 594 | ThermoFisher, Illkirch, France | Cat# S11227 | (1:50) DNA combing |
| Sequence-based reagent | *yap*-MO | This paper | Morpholinos<br>Gene Tools, LLC | 5'TAGGAGACTGTGPGTCACTTCACC 3' |
| Sequence-based reagent | *rif1*-MO | This paper | Morpholinos<br>Gene Tools, LLC | 5'AATCCACAGAACAGACGACAGCCAT 3' |
| Sequence-based reagent | *Control-MO* | This paper | Morpholinos control (Gene Tools Standard Control) Gene Tools, LLC | 5'CCTCTTACCTCAGTTACAATTTATA 3' |
| Recombinant DNA reagent | HLTV-hTRIM21 | Gift from Leo James | RRID: Addgene_104973<br>doi:10.1038/s41596-018-0028-3 | Protein expression |
| Recombinant DNA reagent | *Xenopus rif1* C-Terminal cloned in pET30a vector | Gift from A. Kumagai and W. Dunphy | doi:10.4161/cc.11.6.19636 | Protein expression |
| Recombinant DNA reagent | His-tagged *Xenopus* Yap cloned in pFastBac1vector | Invitrogen | baculovirus Bac-to-Bac expression system Cat# 10359016 | Protein expression |

## Embryo, tadpole, and eye collection

*Xenopus laevis* embryos were obtained by conventional methods of hormone-induced egg laying and in vitro fertilization (*Sive et al., 2007*), staged according to Nieuwkoop and Faber's table of development (*Nieuwkoop et al., 1994*), and raised at 18–20°C. Before whole eye dissection, tadpoles were anesthetized in 0.005% benzocaine. Dissected eye area was measured using AxioVision REL 7.8 software (Zeiss).

## Antibodies and recombinant proteins

A detailed list of the antibodies used in this study for immunohistochemistry (IHC), immunodepletion and western blot (WB) is provided in the Key Resources Table. HLTV-hTRIM21 was a gift from Leo James. Recombinant His-geminin, and His-hTRIM21 were prepared as described respectively (*Clift et al., 2017*; *Toyoshima and Hunter, 1994*). C-terminal *Xenopus rif1* cloned in pET30a vector (a gift from W. Dunphy and A. Kumagai, *Kumar et al., 2012*), was expressed in *Escherichia coli* C41 cells,

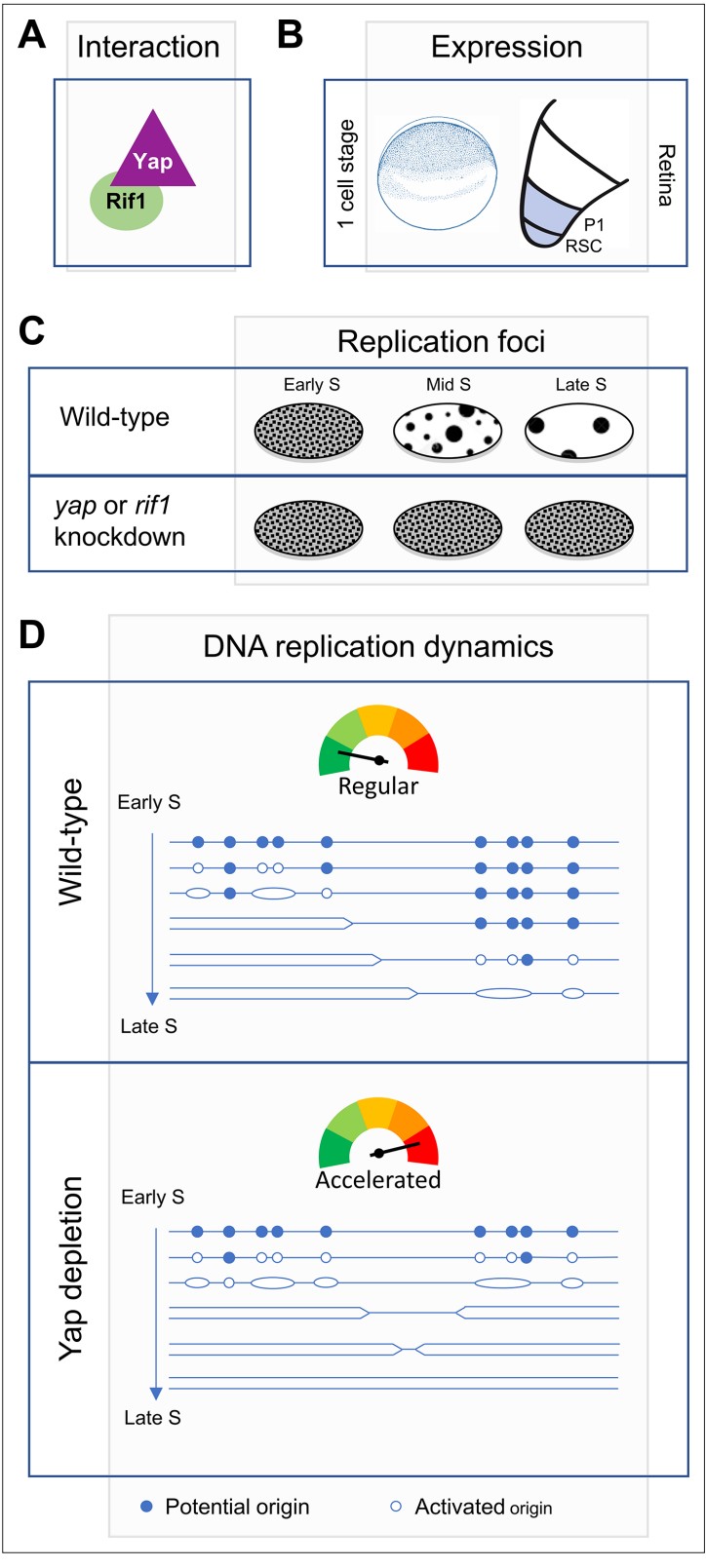

**Figure 8.** Diagram illustrating Yap function in the control of DNA replication dynamics. We found that Yap and Rif1 can interact (**A**) and are co-expressed in *Xenopus* early embryos as well as in retinal stem (RSC)/progenitor cells (P1) (**B**). (**C**) We found that *yap* and *rif1* knockdowns in retinal stem/progenitor cells similarly alter the proper repartition of early and late-like patterns of replication foci (this study and *Cabochette et al., 2015*). (**D**) We

*Figure 8 continued on next page*

*Figure 8 continued*

propose a model where Yap and Rif1 would ensure the proper orchestration of the RT program during early development. The schematic representation of the replication program was adapted from Gaboriaud J and Wu PJ (***Gaboriaud and Wu, 2019***). Based on our assays in vitro in egg extracts and in vivo in early embryos, we propose that following Yap depletion (bottom panel), the number of firing origins is increased, and S-phase length is reduced compared to a wild-type situation (top panel).

purified by Nickel-Sepharose chromatography (Amersham Bioscience), and used as an antigen to raise antibodies in rabbits at a commercial facility (Covalab, Villeurbanne, France). A cDNA encoding recombinant His-tagged *Xenopus* Yap (from the L subgenome) was cloned in pFastBac1vector, expressed in the baculovirus Bac-to-Bac expression system (Invitrogen), purified by Nickel-Sepharose chromatography as described by the supplier (Amersham Bioscience) and then dialyzed overnight against 25 mM Hepes pH 7.8, 250 mM NaCl, 5 mM imidazole, 5% glycerol, 7.5 mM MgCl2, 1 mM DTT, 1 mM EDTA. Purified His-Yap was then used as an antigen to raise antibodies in rabbits at a commercial facility (Covalab, Villeurbanne, France).

## Microinjections

Embryos were injected at the one-cell stage with different components (MO, mRNA, etc.) along with a fluorescent tracer (dextran fluorescein lysine, Thermo Fisher Scientific) to ascertain the correctness of the injections. A total of 200 pg of mRNA (synthesized with mMessage mMachine kit, Life Technologies) were injected, corresponding to the coding region of Yap (FJ979828), Rif1(NM_001280649.1) or GFP as a control. For in vivo depletion experiments, 2 pmol of *Yap*-Morpholinos (MO, Gene Tools, LLC) or 1 pmol of *rif1*-MO or 2 pmol of standard control MO together were microinjected into one-cell stage embryos. The Trim-Away experiments were conducted in a similar way using a mixture of recombinant hTRIM21, anti-Rif1 or anti-Yap antibodies together with 1 or 2 pmol of *rif1*-, *Yap*-, or control-MO (Gene Tools, LLC). MO sequences used in this study can be found in Key Resources Table.

## Replication of sperm nuclei in *Xenopus* egg extracts

Replication competent extracts from unfertilized *Xenopus* eggs and sperm nuclei from testis of male frogs were prepared as described (***Blow and Laskey, 1986***). Sperm nuclei (2000 nuclei/µl) were incubated in untreated, mock, Yap or Rif1 depleted extracts in the presence of cycloheximide to inhibit translation (250 µg/ml, Sigma), energy mix (7.5 mM creatine phosphate, 1 mM ATP, 0.1 mM EGTA, pH 7.7, 1 mM MgCl$_2$). Loading of the MCM complex (pre-RC assembly) was prevented by the addition of 100 nM of recombinant geminin to the extracts.

## Immunodepletions

Rabbit anti-Yap antibody (ab62752, Abcam) or rabbit anti-Rif1 antibody (Covalab, custom-made antibody), pre-immune serum or rabbit IgG (M7023, Sigma) were coupled overnight at 4 °C to protein A Sepharose beads (GE Healthcare). Coupled beads were washed three times in EB buffer (50 mM Hepes, pH 7.5, 50 mM KCl, 5 mM MgCl$_2$). For Yap depletions, coupled beads were then incubated 1 hr at 4 °C in egg extracts (volume ratio 1:3). For Rif1 depletions, coupled beads were incubated 30 min at 4 °C in egg extracts (ratio 1:1) and egg extracts after a first round were re-incubated another 30 min at 4 °C with coupled beads.

## Replication analysis by fluorescence microscopy

Sperm nuclei (2000/µl) were added to replication reactions in the presence of 20 µM rhodamine-dUTP (Roche). At each time point, 20 µl aliquots were diluted in 500 µl PBS and fixed by the addition of 500 µl paraformaldehyde 8%. Nuclei were spun onto coverslips through 1 ml of 20% sucrose cushion in PBS, and counterstained with Hoechst 33,258 as previously described (***Jackson et al., 1995***; ***Labit et al., 2008***). Coverslips were imaged for Hoechst and rhodamine using Olympus BX63 fluorescence microscope to quantify the extent of the rhodamine-dUTP incorporation into DNA. Using Analyze Particles of the Fiji software (***Schindelin et al., 2012***), areas of Hoechst stained nuclei were saved as ROI (region of interest). The rhodamine staining intensity of each ROI was measured as well as the background of each slide. For each nuclei, Corrected Total Fluorescence (CTF) was calculated using the following equations: CTF = Integrated Density of selected nuclei - (Area of selected nuclei X Mean fluorescence of background) (***Gavet and Pines, 2010***).

## Bulk DNA synthesis and alkaline agarose gel electrophoresis

Sperm nuclei (2000 nuclei/µl) were incubated in Mock or Yap depleted extracts and one-fiftieth volume of [α-$^{32}$P]dCTP (3000 Ci/mmol) and reactions were stopped at indicated time points. DNA was recovered after DNAzol treatment (Invitrogen protocol) followed by ethanol precipitation and specific incorporation was measured in cpm in a liquid scintillation analyzer (Tri-Carb4910 TR, Perkin Elmer). Total DNA synthesis (ng/µl) was calculated as described (*Gillespie et al., 2012*). For nascent strand analysis, DNA was separated on 1.1% alkaline agarose gels, and analysed as described (*Marheineke and Hyrien, 2001*). From one extract to another, the replication extent (percent of replication) differs at a specific time point, because each egg extract replicates nuclei with its own replication kinetics. To compare different independent experiments performed using different egg extracts, the data points of each sample were normalized to maximum incorporation value. To include statistics, the scaled data points were grouped into 4 bins (0–25%=early; 26–50%=mid; 51–75%=late; 76–100%=very late S phase); mean and standard deviation were calculated for each bin and a Wilcoxon signed ranked test was used to assess statistically significant differences between the data in each bin.

## Western blot

For analysis of chromatin-bound proteins, we used a protocol slightly modified from *Räschle et al., 2008*. Briefly, reactions were diluted into a 13-fold volume of ELB buffer (10 mM Hepes pH 7.5, 50 mM KCl, 2.5 mM MgCl$_2$) containing 1 mM DTT, 0.2% Triton X100, protease inhibitors and phosphatase inhibitors. Chromatin was recovered through a 500 mM sucrose cushion in ELB buffer at 6780 g for 50 s at 4 °C, washed twice with 200 µl of 250 mM sucrose in ELB buffer, and resuspended in 20 µl SDS sample buffer. Western blots were conducted using standard procedures on *Xenopus* embryo/tadpole protein extracts. Proteins were loaded, separated by 7.5%, 12%, or 4–15% SDS-polyacrylamide gels (Bio-Rad) and transferred into nitrocellulose or ImmobilonP membranes. Membranes were subsequently incubated with the indicated primary antibodies followed by the appropriate horseradish peroxidase-labelled antibodies (1/10,000, Sigma-Aldrich or GE Healthcare, see Key Resources Table). Immunodetection was performed using Super Signal West Pico or Femto Chemiluminescence Kit (Pierce). Quantification was done using Fiji software (*Schindelin et al., 2012*) or using Biorad ImageLab software.

## Molecular combing and detection by fluorescent antibodies

Sperm nuclei were incubated in control-, Yap-, or Rif1-depleted egg extracts in the presence of biotin-dUTP, replication reactions were stopped at indicated time points, DNA was extracted and combed as described (*Marheineke et al., 2009*). Biotin was detected with AlexaFluor594 conjugated streptavidin followed by anti-avidin biotinylated antibodies. This was repeated twice, then followed by mouse anti-human ssDNA antibody, AlexaFluor488 rabbit anti-mouse, and AlexaFluor488 goat anti-rabbit for enhancement (*Gaggioli et al., 2013*). Images of the combed DNA molecules were acquired and measured as described (*Marheineke et al., 2009*). The fields of view were chosen at random. Several hundred of DNA fibers were analysed for each experiment. Measurements on each molecule were made using Fiji software (*Schindelin et al., 2012*) and compiled using macros in Microsoft Excel. Replication eyes were defined as the incorporation tracks of biotin–dUTP. Replication eyes were considered to be the products of two replication forks, incorporation tracks at the extremities of DNA fibers were considered to be the products of one replication fork. Tracts of biotin-labelled DNA needed to be at least 1 kb to be considered significant and scored as eyes. When the label was discontinuous, the tract of unlabelled DNA needed to be at least 1 kb to be considered a real gap. The replication extent was determined as the sum of eye lengths divided by the total DNA length. Fork density was calculated as the total DNA divided by the total number of forks. The midpoints of replication eyes were defined as the origins of replication. Eye-to-eye distances (ETED), also known as inter-origin distances, were measured between the midpoints of adjacent replication eyes. Incorporation tracks at the extremities of DNA fibers were not regarded as replication eyes but were included in the determination of the replication extent or replicated fraction, calculated as the sum of all eye lengths (EL) divided by total DNA. Scatter plots of ETED and EL were obtained using GraphPad version 6.0 (La Jolla, CA, USA). Statistical analyses of repeated experiments have been included as means or medians including standard deviations or ranks as indicated in the legends. A *P*-value ≤0.05 was considered significant.

## Immunostaining and EdU labelling

For immunostaining, tadpoles were anesthetized in 0.005% benzocaine (Sigma), fixed in 1 X PBS, 4% paraformaldehyde 1 hr at room temperature and dehydrated, then embedded in paraffin and sectioned (12 µm) with a Microm HM 340E microtome (Thermo Scientific). Immunostaining on retinal sections was performed using standard procedures. For proliferative cell labelling, tadpoles were injected intra-abdominally, 1 hr prior to fixation, with 50–100 nl of 1 mM 5-ethynyl-2'-deoxyuridine (EdU, Invitrogen) at stage 41. EdU incorporation was detected on paraffin sections using the Click-iT EdU Imaging Kit according to the manufacturer's recommendations (Invitrogen).

Fluorescent images were taken with the AxioImagerM2 with Apotome (Zeiss) coupled to digital AxiocamMRc camera (Zeiss) and processed with the Axio Vision REL 7.8 (Zeiss) and Adobe Photoshop CS4 (Adobe) software. For quantifications of labelled cells by manual cell counting in the CMZ, a minimum of 16 retinas were analysed. Fiji (National Institutes of Health, *Schindelin et al., 2012*) was used to quantify stained areas in the CMZ. All experiments were performed at least in duplicate.

## Co-Immunoprecipitation

Immunoprecipitations from HEK293T cells expressing either HA- or FLAG-tagged Yap (*Cabochette et al., 2015*) were performed using the Dynabeads Protein A Immunoprecipitation Kit (Invitrogen) by coupling 5 µg of anti-FLAG (Cell signaling) to the beads and following the manufacturer's protocol. Immunoprecipitations from *Xenopus* egg extracts were performed as described below for mass spectrometry using rabbit anti-Yap (ab62752, Abcam) or rabbit anti-Rif1 antibodies. The 7.5% polyacryl-amide gel was further analysed by western blot using Mouse anti-Yap or rabbit anti-RIF antibodies.

Antibodies used for immunoprecipitation are listed in Key Resources Table.

## Mass spectrometry

Rabbit anti-Yap antibody (ab62752, Abcam) or rabbit IgG (M7023, Sigma) were coupled 2 hr at RT to protein A Sepharose beads (GE Healthcare). Coupled beads were covalently crosslinked using dimethyl pimelimi-date according to standard procedures, washed with PBS and kept in PBS, 0.02% sodium azide at 4 °C. For IP experiments, crosslinked beads with rabbit anti-Yap antibody or rabbit IgG were washed three times in EB buffer (50 mM Hepes, pH 7.5, 50 mM KCl, 5 mM MgCl$_2$) and were incubated in *Xenopus* egg extracts for 30 min at 4 °C. Beads were isolated by centrifugation, washed three times with EB buffer then once in EB buffer, 0.01% Tween 20. The immunoprecipitated proteins were eluted by 2 X Laemmli buffer and collected after centrifugation. Approximately 20 ng of immunoprecipitated Yap protein fraction was loaded on a 7.5% polyacrylamide gel and analysed by mass spectrometry (Protéomique Paris Saclay-CICaPS platform). Protein samples were reconstituted in solvent A (water/ACN [98: 2 v/v] with 0.1% formic acid) and separated using a C18-PepMap column (Thermo Fisher Scientific) with a solvent gradient of 2–100% Buffer B (0.1% formic acid and 98% acetonitrile) in Buffer A at a flow rate of 0.3 µl/min. The peptides were electrosprayed using a nanoelectrospray ionization source at an ion spray voltage of 2300 eV and analysed by a NanoLC-ESI-Triple TOF 5600 system (AB Sciex). Protein identification was based on a threshold protein score of >1.0. For quan-titation, at least two unique peptides with 95% confidence and a p-value <0.05 were required.

Comprehensive protein list analysis and enriched biological pathways were based on Gene ontology classification system using Metascape (*Sajgo et al., 2017*). Data visualization was done using GOPlot R package (*Livak and Schmittgen, 2001*).

## Quantification and statistical analyses

For quantifications of labelled EdU$^+$ cells by manual cell counting in the CMZ, 16–11 retinas per condition with a minimum of two sections per retina were analysed. Dissected eye areas and the number of cells per embryo were measured using Adobe Photoshop CS4 software. All experiments were performed at least in duplicate. Shown in figures are results from one representative experiment unless specified.

Statistical analyses (GraphPad Prism software, version 8.3.0) were performed using a Mann-Whitney test or Wilcoxon signed ranked test as mentioned in the figure legends.

## Acknowledgements

We are thankful to A Kumagai and B Dunphy for the gift of the C-terminal *Xenopus* Rif1 containing plasmid and *Xenopus* Rif1 antibody. We are grateful to A Donval for her help with frog fertilizations

and Virginie Chiodelli for technical support. This research was supported by grants to M.P. from ARC (Association pour la Recherche sur le Cancer), Association Retina France, Fondation Valentin Haüy, and UNADEV (Union Nationale des Aveugles et Déficients Visuels) in partnership with ITMO NNP (Institut Thématique Multi-Organisme Neurosciences, sciences cognitives, neurologie, psychiatrie) / AVIESAN (alliance nationale pour les sciences de la vie et de la santé). RMG was a Conacyt fellow (grant number 439641). This work has benefited from the facilities and expertise of the I2BC proteomic platform (Proteomic-Gif, SICaPS) supported by IBiSA, Ile de France Region, Plan Cancer, CNRS and Paris-Saclay University.

## Additional information

### Funding

| Funder | Grant reference number | Author |
|---|---|---|
| Association pour la Recherche sur le Cancer | | Muriel Perron<br>Odile Bronchain |
| Retina France | | Muriel Perron |
| Fondation Valentin Haüy | | Muriel Perron |
| UNADEV | | Muriel Perron |
| Conacyt | 439641 | Rodrigo Meléndez García |

The funders had no role in study design, data collection and interpretation, or the decision to submit the work for publication.

### Author contributions

Rodrigo Meléndez García, Resources, Formal analysis, Validation, Investigation, Visualization, Methodology, Writing - original draft; Olivier Haccard, Resources, Formal analysis, Validation, Investigation, Visualization, Methodology, Writing - original draft, Writing - review and editing; Albert Chesneau, Resources, Formal analysis, Validation, Investigation, Visualization, Methodology; Hemalatha Narassimprakash, Resources, Validation; Jérôme Roger, Conceptualization, Resources, Formal analysis, Validation, Investigation, Visualization, Writing - original draft, Writing - review and editing; Muriel Perron, Conceptualization, Formal analysis, Supervision, Funding acquisition, Validation, Visualization, Methodology, Writing - original draft, Project administration, Writing - review and editing; Kathrin Marheineke, Odile Bronchain, Conceptualization, Resources, Formal analysis, Supervision, Funding acquisition, Validation, Investigation, Visualization, Methodology, Writing - original draft, Project administration, Writing - review and editing

### Author ORCIDs

Olivier Haccard (ID) http://orcid.org/0000-0002-4305-2746
Muriel Perron (ID) http://orcid.org/0000-0002-1558-8236
Kathrin Marheineke (ID) http://orcid.org/0000-0002-1514-0167
Odile Bronchain (ID) http://orcid.org/0000-0001-8932-8907

### Ethics

All animal experiments have been carried out in accordance with the European Community Council Directive of 22 September 2010 (2010/63/EEC). All animal care and experimentation were conducted in accordance with institutional guidelines, under the institutional license C 91-471-102. The study protocols were approved by the institutional animal care committee CEEA #59 and received an authorization from the Direction Départementale de la Protection des Populations under the reference APAFIS#998-2015062510022908v2 for Xenopus experiments.

### Decision letter and Author response

Decision letter https://doi.org/10.7554/eLife.75741.sa1
Author response https://doi.org/10.7554/eLife.75741.sa2

## Additional files

### Supplementary files

• Supplementary file 1. Depletion of Yap increases replication origin firing in *Xenopus* egg extracts. Extended DNA combing data to *Figure 2* of 2 independent experiments, Replicate 1 and Replicate 2.

• Supplementary file 2. Depletion of Rif1 increases replication origin firing in *Xenopus* egg extracts. Extended DNA combing data to *Figure 4* of 2 independent experiments, Replicate 1 and Replicate 2.

• Transparent reporting form

### Data availability

Source data files have been provided for all Western blots and graphs shown on the figures. We have submitted our dataset "Identification of Yap-interacting proteins in *Xenopus* egg extracts by co-immunoprecipitation coupled to LC/MS/MS" to ProteomeXchange via the PRIDE database (Project accession: PXD029345; Project DOI: https://doi.org/10.6019/PXD029345).

The following dataset was generated:

| Author(s) | Year | Dataset title | Dataset URL | Database and Identifier |
|---|---|---|---|---|
| Meléndez García R, Haccard O, Chesneau A, Narassimprakash H, Roger JE, Perron M, Marheineke K, Bronchain O | 2021 | Identification of Yap-interacting proteins in Xenopus egg extracts by co-immunoprecipitation coupled to LC/MS/MS | https://www.ebi.ac.uk/pride/archive/projects/PXD029345 | PRIDE, PXD029345 |

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
