## [Editor Report]

The YAP protein is well-known as a major regulator of tissue growth and repair, acting as a co-factor of transcription to promote cell proliferation and de-differentiation. In *Xenopus* egg extracts and embryos, before the initiation of transcription, this manuscript now elegantly identifies a new role of YAP in DNA replication dynamics, thus slowing down cell division rate.

---

## [Decision Letter]

**Decision letter after peer review:**

Thank you for submitting your article "A non-transcriptional function of Yap orchestrates the DNA replication program" for consideration by *eLife*. Your article has been reviewed by 3 peer reviewers, and the evaluation has been overseen by a Reviewing Editor and Jessica Tyler as the Senior Editor. The reviewers have opted to remain anonymous.

Essential revisions:

1) The quantification of the amount of YAP in Figure 1B is confusing. The legend of the chart states "Control in light grey and presence of geminin in black", but the bar colors are of different shades of grey. It is not clear how to evaluate them. The graph should be improved by properly labeling the legend of the figure or, simply, by showing the relative amount of YAP intensity values relative to a loading control like H3, normalized to the 90 min control condition.

The higher rate of DNA synthesis observed in the absence of Yap in Figure 1D is not very evident from the gels in Figure 1- supplement 3B. The timing of the experiments is continuously changing throughout the figures. It is therefore difficult to compare them. Also, comparisons across different gels are difficult to interpret.

Most importantly, relative quantification on gel images cannot support the claim of increased DNA synthesis in the absence of YAP. To accurately quantify the replication of DNA added to the extract, the total amount of DNA synthesized must be quantified. This can be easily done by counting labeled nucleotide incorporation on precipitated DNA (TCA replication assay and other methods have been described by Mechali and Blow labs; see Gillespie et al., Methods Volume 57, Issue 2, June 2012, Pages 203-213). Without these data, the central claim of the paper is meaningless.

It is necessary to analyze the dynamics and the abundance of chromatin-bound replication proteins associated with the active replication fork after Yap depletion using chromatin binding assays. This would further confirm the increase in the fork density observed by DNA combing experiments.

2) Why are only 2 replicates instead of 3 provided in Figure 2? This is especially true for the conclusion that eye length is unaltered -it appears that there is a subset of eye length that is increased in 2F, which might reach significance if triplicates were performed.

3) The authors suggest that YAP and RIF1 work together, – it would be interesting to co-deplete YAP and RIF1. If YAP works solely through Rif1, it should not be more effective than loss of YAP alone.

Does loss of RIF1 affect YAP loading onto chromatin?

The efficiency of depletion for both RIF1 and YAP is different in Figure 4B and Figure 4A, supplement 1. Moreover, the combined use of the TRIM-away approach with injections of MO led to a stronger and prolonged YAP depletion but also triggered toxicity in the tadpoles, which display severe abnormalities. Therefore, the outcomes of these procedures are unlikely to be specific, unless the authors can provide a rescue experiment. Quantitation of the Western blots in panel A of Figure 4 supplement 1 is needed to be convincing that trim away and morpholino combined are more effective.

4) It is difficult to see the points the authors wish to communicate in Figure 6. If the authors add yellow and red arrows in B to C and D, this might make it clearer. There is almost no EdU in the YAP-MO -does this affect the ability to recognize the different patterns in this region of the eye?

5) The title of the manuscript is "A non-transcriptional function of YAP orchestrates the DNA replication program". It is not clear that YAP "orchestrates" DNA replication – for this to be true, it would have to be signal responsive. Since the authors did not reveal any links to YAP activity (such as YAP phosphorylation or nuclear/cytoplasmic distribution) it is not "orchestrating" DNA replication. Please adjust the title accordingly and also provide a clear indication of the biological system under investigation.

6) Statistics

Please indicate the statistical test in the legend of Figures 1B, 2E-F, 5C.

---

## [Author Response]

Essential revisions:1) The quantification of the amount of YAP in Figure 1B is confusing. The legend of the chart states "Control in light grey and presence of geminin in black", but the bar colors are of different shades of grey. It is not clear how to evaluate them. The graph should be improved by properly labeling the legend of the figure or, simply, by showing the relative amount of YAP intensity values relative to a loading control like H3, normalized to the 90 min control condition.

We apologize for this confusion. This has been corrected and the Figure 1B is now properly labelled.

The higher rate of DNA synthesis observed in the absence of Yap in Figure 1D is not very evident from the gels in Figure 1- supplement 3B.

Since Figure 1- supplement 3B was not clear enough, we replaced this image of the gel electrophoresis of nascent strands by another one, now displayed in Figure 1—figure supplement 2. We think that this new figure better illustrates both the quantitative aspect of replication differences in terms of raw intensities measured of whole lanes on gels but also the qualitative view of replication. We think that it is important for the readers to visualize and to be able to compare the length of nascent strains between Mock- and Yap-depleted extracts, which is also a proof of experimental quality of the DNA synthesis in vitro, which is not visible in absolute quantification after scintillation counting.

The timing of the experiments is continuously changing throughout the figures. It is therefore difficult to compare them. Also, comparisons across different gels are difficult to interpret.

The referee is correct but as explained in the MM section page 14 of the original manuscript, the replication extent (percent of replication) differs for a specific time point from one extract to another, because each egg extract prepared from one batch of eggs from one frog replicates nuclei with its own replication kinetics. To overcome this problem and to compare different independent experiments performed using different egg extracts, the data points of each sample were normalized to maximum incorporation value.

Most importantly, relative quantification on gel images cannot support the claim of increased DNA synthesis in the absence of YAP. To accurately quantify the replication of DNA added to the extract, the total amount of DNA synthesized must be quantified. This can be easily done by counting labeled nucleotide incorporation on precipitated DNA (TCA replication assay and other methods have been described by Mechali and Blow labs; see Gillespie et al., Methods Volume 57, Issue 2, June 2012, Pages 203-213). Without these data, the central claim of the paper is meaningless.

Although we do not agree that relative quantification on gel images cannot support the claim of increased DNA synthesis in the absence of Yap, we thank the reviewer for his suggestion since we now provide additional data clearly strengthening our conclusion.

Many studies, published in high standards journals and coming from different *Xenopus* replication laboratories have quantified DNA synthesised after ^32^P-dCTP incorporation followed by separation by agarose gel electrophoresis using percentage of maximal incorporation or arbitrary units (Shechter *et al.,* 2004; Trenz *et al.,* 2008; Guo *et al.,* 2015; Walter and Newport, 1997; Suski et al., 2022, Nature). Nevertheless, as the referee suggested, we quantified the total amount of DNA synthesized in three new independent experiments. These new results, presented page 5, lines 34-39 and shown in Figure 1G, support our conclusion, as they also show that Yap depletion increases total DNA synthesis. Please note that the DNA combing results presented in Figure 2 already showed that replication is increased after Yap depletion when quantifying total replication percentages of all combed DNA. Finally, we also added another set of experiments to Figure 1 to further confirm these findings. We used the incorporation of rhodamine-dUTP followed by the quantification of the fluorescence intensity within nuclei. This nuclei-fluorescence based method is frequently used in proliferation assays to assess nucleotide incorporation resulting from the DNA replication process in other organisms. Our new results demonstrate that DNA synthesis is increased 1.5-fold in six biological replicates and represent a third independent method, in addition to DNA combing and ^32^P-dCTP incorporation, showing that DNA synthesis is increased upon YAP depletion. These new results are now presented page 5, lines 2734 and shown in Figure 1D-F.

Considering all these additional data, we had to move and replace the alkaline gel from the original Figure 1D to Figure 1—figure supplement 2.

It is necessary to analyze the dynamics and the abundance of chromatin-bound replication proteins associated with the active replication fork after Yap depletion using chromatin binding assays. This would further confirm the increase in the fork density observed by DNA combing experiments.

We thank the referee for this suggestion and we added a western blot of chromatin bound proteins after Yap depletion. This shows that two replication proteins associated with the active replication fork, namely Cdc45 and PCNA, are enriched after Yap depletion compared to the control at the beginning of S-phase. This observation further supports the DNA combing results showing that more forks are active after YAP depletion. This new data is now presented page 6, lines 25-32 and displayed in Figure 2H.

We would like to stress here that with these additional methods added to the revised version, five different methods in total (Rhodamine-dUTP incorporation/nucleus, 32P-dCTP incorporation – total synthesis, 32P-dCTP incorporation – nascent strand analysis, DNA combing, western blotting for replication fork proteins) show that DNA synthesis and origin activation is increased after Yap depletion.

2) Why are only 2 replicates instead of 3 provided in Figure 2?

Answer: We apologize for this confusing presentation. In Figure 2A, three fibers were shown as visual examples illustrating the results of this experiment (and not one fiber per replicate). We have two replicates with two different time points, as stated in the text each, and shown in the tables. We modified the legend of the figure to make this clearer*.*

We also changed the fibers images (Figure 2A) to better illustrate the data set (see point below).

This is especially true for the conclusion that eye length is unaltered -it appears that there is a subset of eye length that is increased in 2F, which might reach significance if triplicates were performed.

The scatter plot in Figure 2F makes it look like that there are more eyes with larger sizes after Yap depletion, but please note that there are also more EL measured as stated in the legend (ΔMock n=182 versus ΔYap n=311). To highlight this important parameter, we added these numbers below the scatter plot in the revised Figure 2F, as we have done consistently for all of the experiments presented in the revised Figures. The means of these two EL distributions are indeed numerically different but since both distributions are not Gaussian (d'Agostino and Pearson test), only non-parametric tests can apply (Mann-Whitney or Kolmogorov Smirnow test). The results of the two non-parametric tests showed that the distributions are not significantly different as mentioned in the legend. However, we cannot rule out that after Yap depletion, some larger eyes may arise from fusions of forks or from a higher fork speed, but again, the tests applied to a high number of measurements show no significant statistical differences. For later time points, this issue is discussed in the original manuscript p9 and still now in the revised manuscript page 6, line 22-25.

3) The authors suggest that YAP and RIF1 work together, – it would be interesting to co-deplete YAP and RIF1. If YAP works solely through Rif1, it should not be more effective than loss of YAP alone.

Double immunodepletions are technically very challenging. We did not manage to efficiently co-deplete Yap and Rif1 from egg extracts while maintaining a sufficient replication efficiency. We did however directly compare the effects of YAP depletion to those of Rif1 depletion alone. As for Yap depletion, we first quantified rhodamine-dUTP incorporation after Rif1 depletion by direct fluorescence microscopy that demonstrated a clear increase of DNA synthesis, consistent with Alver et al., 2017. Second, we performed 2 independent DNA combing experiments after Rif1 depletion with a large number of fibers analyzed (total 10 000) in egg extracts that show a marked increase in DNA replication and fork density like those seen after Yap depletion, spanning from very early to mid Sphase. Please note that this is also the first replication kinetics analysis by DNA combing after Rif1 depletion in the *Xenopus* in vitro system which clearly shows that Rif1 depletion leads to a strong increase of origin activation during early S-phase. We therefore found that Rif1 depletion and Yap depletion qualitatively show the same main effects: an increase of DNA synthesis and fork density, that are more pronounced in early S-phase. We also noticed quantitative differences in the direct fluorescence after rhodamine incorporation of whole nuclei and fork density, with stronger effects after Rif1 depletion compared to Yap depletion. This suggests that there might be an additional mechanism for Rif1 in regulating origin activation. These new data, presented page 7, lines 4-29, have been added in the form of an entire new Figure 4.

Does loss of RIF1 affect YAP loading onto chromatin?

We thank the referee for this interesting question, which we addressed already after the preprint deposition of our manuscript on BioRxiv upon a request of a reader. We could not observe any major changes in Yap chromatin recruitment after Rif1 depletion nor the reverse in our conditions (Figure 1). However, since after Rif1 depletion we can still detect some residual Rif1 left on chromatin, we cannot completely exclude that this residual Rif1 is sufficient to recruit Yap. We were unable to further improve Rif1 depletion without severely compromising the replication capacity of the Rif1depleted egg extract. This difficulty could be explained by the abundance of Rif1 and/or its binding to insoluble nuclear structure. Since the result is not 100 % conclusive for us to be published, we prefer not to include it into the manuscript and we show the blots for information to the referees, see Author response image 1.

**Author response image 1. sa2fig1:** Rif1 depletion (A) or Yap depletion (B) does not inhibit Yap or Rif1 recruitment, respectively, onto chromatin in egg extracts.

The efficiency of depletion for both RIF1 and YAP is different in Figure 4B and Figure 4A, supplement 1.

We agree with the referee that the efficiency of depletion is different in both figures. This is explained by the fact that the extent of the depletion varies from experiment to experiment. We work with different batches of in vitro fertilized embryos and extracts, so these differences simply reflect the technical/biological variability.

Moreover, the combined use of the TRIM-away approach with injections of MO led to a stronger and prolonged YAP depletion but also triggered toxicity in the tadpoles, which display severe abnormalities. Therefore, the outcomes of these procedures are unlikely to be specific, unless the authors can provide a rescue experiment.

It is important to point out that abnormal development is not always attributable to a toxic effect. Many losses of gene function result in malformations without being ascribed to toxicity or unspecific effects. However, we agree with the reviewers on the need to present a rescue experiment, which is now shown in new Figure 5C and new Figure 5—figure supplement 1B. In addition, we also provide gain-of-function (GOF), data for YAP in early embryos. In brief, we find that the Yap GOF leads to opposite outcomes than those of its depletion with embryos at the same stage of development, having fewer and larger cells than the control. Furthermore, we show that the effects of Yap depletion, i.e. embryos with more and smaller cells than the control at the same developmental stage, are rescued by the addition of mRNA encoding Yap to restore the protein level. This is true for both embryonic divisions (new Figure 5C) and development, as we obtained normal-looking neurula after Yap rescue (new Figure 5—figure supplement 1C). Overall, these data now clearly show that Yap is both sufficient and necessary to maintain the rate of embryonic divisions and that this phenotype is specific since it can be rescued by expressing Yap alone. These new data are presented pages 8, lines 2-10.

Quantitation of the Western blots in panel A of Figure 4 supplement 1 is needed to be convincing that trim away and morpholino combined are more effective.

The quantification is now presented in new Figure 5—figure supplement 1A. At the 2 -cell stage, we observe some fluctuations in the amounts of Yap between samples, the origin of which we do not fully understand. At the 4-cell stage, a reduction in Yap is observed regardless of the depletion strategy used. It is from the 8-cell stage onwards that differential effects between the depletion methods can be appreciated. From this point, the quantifications confirm that the TRIM-away and Morpholino combined are more effective than taken separately with regard to Yap (indicated by green arrows in Figure).

4) It is difficult to see the points the authors wish to communicate in Figure 6. If the authors add yellow and red arrows in B to C and D, this might make it clearer.

We thank the reviewer for this suggestion. We have added different coloured stars according to the replication patterns directly on the nuclei observed on the CMZ enlargements (new Figure 7BC). This representation makes it possible to immediately identify the different patterns without obscuring the image.

There is almost no EdU in the YAP-MO -does this affect the ability to recognize the different patterns in this region of the eye?

Our observations show that there are fewer EdU positive cells in the Yap-MO but not “no EdU”. The fluorescence intensity in the green-labelled nuclei in Figure 7 C after Yap MO does not appear different from that in the control-MO. Under these conditions, there is no reason to think that one pattern is more difficult to recognise than the other one.

5) The title of the manuscript is "A non-transcriptional function of YAP orchestrates the DNA replication program". It is not clear that YAP "orchestrates" DNA replication – for this to be true, it would have to be signal responsive. Since the authors did not reveal any links to YAP activity (such as YAP phosphorylation or nuclear/cytoplasmic distribution) it is not "orchestrating" DNA replication. Please adjust the title accordingly and also provide a clear indication of the biological system under investigation.

By “orchestrate”, we had in mind an actor able to modulate the speed of execution of the replication process. Nonetheless, we acknowledge that this may be open to interpretation. We thus propose to replace “orchestrates” by “regulates”.

6) StatisticsPlease indicate the statistical test in the legend of Figures1B, 2E-F, 5C.

We have now added the statistical tests in the legends for these Figures.